# Closing the gap in the tropics: the added value of radio-occultation data for wind field monitoring across the equator

Julia Danzer[1], Magdalena Pieler[1], and Gottfried Kirchengast[1,2]

[1]Wegener Center for Climate and Global Change, University of Graz, Graz, 8010, Austria
[2]Institute of Physics, University of Graz, Graz, 8010, Austria

**Correspondence:** Julia Danzer (julia.danzer@uni-graz.at)

**Abstract.** Globally available and highly vertically resolved wind fields are crucial for the analysis of atmospheric dynamics for the benefit of climate studies. Most observation techniques have problems to fulfill these requirements. Especially in the tropics and in the southern hemisphere more wind data are required. In this study we investigate the potential of radio occultation (RO) data for climate-oriented wind field monitoring in the tropics, with a specific focus on the equatorial band within $\pm 5°$ latitude. In this region, the geostrophic balance breaks down, due to the Coriolis force term approaching zero, and the equatorial balance equation becomes relevant. One aim is to understand how the individual wind components of the geostrophic balance and equatorial balance approximations bridge across the equator and where each component breaks down. Our central aim focuses on the equatorial balance approximation, testing its quality by comparison with ERA5 reanalysis data. The analysis of the zonal and meridional wind component showed that while the zonal wind was well reconstructed, it was difficult to estimate the meridional wind from the approximation. However, we still found a somewhat better agreement from including both components in the zonal-mean total wind speed in the troposphere. In the stratosphere, the meridional wind component is close to zero for physical reasons and has no relevant impact on the total wind speed. In general, the equatorial balance approximation works best in the stratosphere. As a second aim we investigated the systematic data bias between using the RO and ERA5 data and find it smaller than the bias resulting from the approximations. We also inspected the monthly-mean RO wind data over the full example year 2009. The bias in the core region of highest quality of RO data, which is the upper troposphere and lower stratosphere, was generally smaller than $\pm 2\,\mathrm{m\,s^{-1}}$. This is in line with the wind field requirements of the World Meteorological Organization. Overall, the study encourages the use of RO wind fields for meso-scale climate monitoring over the entire globe, including the equatorial region, and also showed a small improvement in the troposphere when including the meridional wind component in the zonal-mean total wind speed.

## 1 Introduction

Globally available upper air wind profiling information is crucial for the analysis of atmospheric dynamics for the benefit of climate studies, as well as climate models and numerical weather prediction. To determine a wind flow in its full state, wind-sensitive measurements need to ensure a high, three-dimensional resolution, global coverage, and frequent observations from

the troposphere to the stratosphere (English et al., 2013; Baker et al., 2014; Eyre et al., 2020). However, wind measurements in the free atmosphere, depending on the observing system, lack very often one or more of these requirements.

Stoffelen et al. (2005, 2020) emphasize the need for horizontal resolutions smaller than 10 km to 500 km, to follow an atmospheric process in detail from initial small-scale amplitudes to evolving dynamical mesoscale structures. The World Meteorological Organization (WMO) and the Observing Systems Capability Analysis and Review tool (OSCAR) require a vertical resolution of wind information of about $1\,\mathrm{km}$ in the troposphere and $2\,\mathrm{km}$ in the stratosphere, for weather and climate applications, with a wind accuracy of $2\,\mathrm{m\,s^{-1}}$ (see WMO-OSCAR, 2023).

Furthermore, well-resolved wind data need to be available over the oceans, tropics, and southern hemisphere, where often a measurement gap is present. For example, land surface stations, ships, buoys, and wind scatterometers from satellites provide valuable surface data, but lack vertical profiling information. Aircrafts and atmospheric motion vectors (AMVs) from geostationary or polar satellites provide a high temporal and horizontal sampling at several heights, but have distinct limits in accurate vertical geolocation and resolution and global representation. Wind profilers, radiosondes, and pilot balloons, have a high vertical sampling, but provide information primarily at single locations over continents and the northern hemisphere. On the other hand, the Atmospheric Dynamics Mission (ADM-Aeolus, operating over August 2018 to July 2023) provided 3D wind profiling with a frequent and high-resolution coverage, filling measurement gaps over the oceans, poles, tropics, and the southern hemisphere, up to an altitude of about 20 km. However, it depended on clear-air molecular scattering (no measurements within clouds) and on hydrometeor Mie scattering, which can be particularly tricky at tropical latitudes, due to the high-altitude cloud systems (see also, Stoffelen et al., 2005, 2020; Kanitz et al., 2019). Finally, wind information is nowadays obtained also implicitly as part of variational data assimilation ("4D-Var") in numerical weather prediction analyses that initialize the forecasts, such as through the geostrophic adjustment and directly through the background error covariances (especially where the geostrophic balance applies) as well as through 4D-Var of humidity and/or ozone tracing data (Geer et al., 2018; Zaplotnik et al., 2023).

In this respect, a valuable complementary data source comes from exploiting a different satellite-based observation technique, the Global Navigation Satellite System (GNSS) radio occultation (RO) method. RO provides vertical profiles of geophysical variables such as refractivity, density, pressure, and temperature. A basic introduction to the RO method can be found in, e.g., Kursinski et al. (1997); Hajj et al. (2002). The applications range across climate monitoring and climate analysis, numerical weather prediction, as well as space weather applications (e.g., Healy, 2007; Cucurull, 2010; Foelsche et al., 2009; Anthes, 2011; Steiner et al., 2011).

There are several key advantages of RO data, which could make them a beneficial observation-based data set for (indirect) wind field monitoring. First of all, it provides a multi-satellite, long-term stable, global data set record, with no need for inter-calibration between the missions (Wickert et al., 2001; Anthes et al., 2008; Foelsche et al., 2011a; Angerer et al., 2017; Steiner et al., 2020). In addition, RO provides all-weather capability, which is a specific advantage in the tropics with large high-altitude cloud systems that can limit other observation systems, such as optical sounders. Furthermore, RO data are a high vertical resolution data set, with a resolution of about $100\,\mathrm{m}$ to $200\,\mathrm{m}$ in the troposphere, to about $500\,\mathrm{m}$ in the lower stratosphere at low to mid-latitudes, and near $1.5\,\mathrm{km}$ from the middle stratosphere towards high altitudes (Schwarz et al.,

2017, 2018; Zeng et al., 2019). RO data cover well the (free) troposphere and the stratosphere, with a core region of high quality in the upper troposphere and lower stratosphere (e.g., Zeng et al., 2019; Steiner et al., 2020), having a horizontal resolution of about 200 km to 300 km (e.g., Kursinski et al., 1997; Foelsche et al., 2011b). Hence it can give additional wind profiling information at higher altitude regions, where other observation-based data sets might only cover the troposphere and lower stratosphere (e.g., radiosondes, Ladstädter et al. (2015); Bodeker et al. (2016)).

Traditionally, most RO climate studies concentrate on using the high-quality vertical temperature information (e.g., Li et al., 2023; Ladstädter et al., 2023). With respect to numerical weather prediction, the RO bending angle or refractivity profiles are assimilated in forecasting and reanalysis systems (e.g., Kuo et al., 2000; Cardinali and Healy, 2014; Hersbach et al., 2020). It is important in this respect, emphasized in Scherllin-Pirscher et al. (2017), that RO data have the power of vertical geolocation, meaning they provide accurate information on the absolute altitude of a measured air parcel. Hence, RO provides virtually independent information on altitude and pressure fields, enabling also to study an accurate representation of the mass field driven wind field circulation. So far, only a few studies have analyzed the option of calculating wind fields from RO geopotential fields on isobaric levels. Scherllin-Pirscher et al. (2014) and Verkhoglyadova et al. (2014) have tested the geostrophic wind approximation, excluding the tropics completely between $\pm 15°$ latitude. Healy et al. (2020), on the other hand, tested the zonal equatorial balance equation around the equator, studying the utility of RO data in a 5°-zonal band in the stratosphere.

In a previous study, Nimac et al. (2023) analyzed the geostrophic approximation on a monthly 2.5° x 2.5° latitude x longitude grid for ERA5-reanalysis and RO data. It was possible to reproduce the original ERA5 winds fairly well, and within the target accuracy of $\pm 2\,\mathrm{m\,s^{-1}}$. However, in the region of the jet stream, the difference between the two data sets exceeded this target. Furthermore, over large mountain areas (e.g., Himalayan or Andes region) larger deviations were found, since the ageostrophic contribution grows in importance in such regions with massive influence of topography. Our study furthermore showed that within the 2007 to 2020 evaluation period the difference between RO and ERA5 became noticeably smaller from 2016 onward, coinciding with an ERA5 observing systems change including as of 2016 additional information from various sources such as land stations, ships, and buoys. This emphasized the temporal stability of RO data and also points to the high-quality of RO data (Steiner et al., 2020). In general, the wind speed estimates performed well towards the tropics up to even $\pm 5°$ around the equatorial band. Within the equator band, the Coriolis force approaches zero and the singularity starts to dominate. For this physical reason it is not possible to use the geostrophic approximation to retrieve wind fields over a narrow band around the equator, leaving a gap in RO wind field computation.

In this study we aim to close this gap by deriving RO winds across the equator. While in the important pre-work of Healy et al. (2020) a stratospheric zonal-mean zonal wind field was derived in a 10° equatorial band, we aim to compute latitudinal x longitudinal resolved wind fields with RO data. For this purpose we investigate the zonal ($u$) and meridional ($v$) wind components, as well as total wind speed ($V$), based on the equatorial balance equation (Chandra et al., 1990; Scaife et al., 2000; Holton, 2004). The method and the data sets used are introduced in Section 2 and Section 3. In a first step, we assess the quality of the approximation, using monthly ERA5 reanalysis data (Hersbach et al., 2020) on a 2.5° x 2.5° grid as a reference. Here we compare the original ERA5 wind components and wind speeds to the ones computed from the equatorial

balance approximation (Section 4.1). In a second step, we derive the zonal and meridional wind components, as well as total wind speed, for monthly RO climatologies, analyzing the quality and added value of RO wind field products over the equatorial band (Section 4.2). Finally, in Section 4.3, we test how the equatorial-balanced wind speeds bridge the geostrophic wind speeds across the equator, closing the gap in the tropics with RO wind data. Summary and Conclusions are then given in Section 5.

The overarching goal is to collect the knowledge from the prior (Nimac et al., 2023) and this current study, to produce a long-term stable global climate RO wind field record, covering the upper troposphere up to the middle stratosphere, at monthly and meso-scale resolution. In this respect the added value of RO data can play out; its unique combination of high vertical resolution, accuracy, and long-term stability (=multi-year to multi-decadal stability). The possible applications are numerous, from global climate wind field monitoring up to studies of changes in climate-related wind field dynamics.

## 2 Method for wind field derivation

In general a wind flow in the free atmosphere can be approximated by geostrophic balance, which equals an exact balance between Coriolis force and pressure gradient force. Friction can be ignored in the free atmosphere, while ageostrophic contributions become generally of higher relevance in the winter hemisphere, and also above large mountain areas (see e.g., Scaife et al., 2000; Nimac et al., 2023). The geostrophic balance breaks down in the tropics, due to the Coriolis force approaching zero, inducing a singularity in the geostrophic approximation. A solution for the wind equation in the tropics, assuming a steady friction-less flow, is the equatorial balance equation. In this study we calculate wind speeds, using the geostrophic balance and equatorial balance approximations, with the main focus on the latter one. The derivation of RO wind fields, based on the geostrophic approximation, has already been thoroughly validated in a prior study (Nimac et al., 2023). In our analysis we follow the accuracy requirements specified by the World Meteorological Organization (WMO), see WMO-OSCAR, 2023. The WMO provides detailed and differentiated requirements, for different spatial and temporal resolutions, as well as for different applications (e.g., applications in numerical weather prediction). Since our focus are monthly-averaged meso-scale winds relevant for the description of climate, we use an indicative threshold of wind speed biases smaller than $\pm 2\,\mathrm{m\,s^{-1}}$. We further note that the advantage of RO-based long-term wind records is their unique potential of being temporally stable, which is another WMO requirement of stability. Considering monthly-mean wind speeds with accuracy within $\pm 2\,\mathrm{m\,s^{-1}}$, this is roughly consistent with a decadal stability of $\pm 0.5\,\mathrm{m\,s^{-1}}$ per decade, which is the associated WMO-based requirement that we use to evaluate long-term stability (see Nimac et al., 2023).

**The equatorial balance equation:** to derive wind fields over the equator, we follow the formulation of Chandra et al. (1990); Scaife et al. (2000). The equatorial wind data are derived from geopotential $\Phi$, given on isobaric levels, resulting in the following formulation for the zonal and meridional wind components, $u_{eb}$ and $v_{eb}$, respectively, over the equator:

$$u_{eb} \simeq -\frac{1}{\beta R_E^2}\frac{\partial^2 \Phi}{\partial \varphi^2}\,, \tag{1}$$

$$v_{eb} \simeq \frac{1}{\beta R_E^2}\frac{\partial^2 \Phi}{\partial \varphi \partial \lambda}\,, \tag{2}$$

where $\beta$ equals $2\Omega/R_\mathrm{E}$, with $\Omega$ being the Earth's angular rotation rate ($7.2921 \times 10^{-5}$ rad/s), and $R_\mathrm{E}$ is the Earth's mean radius ( 6371 km). $\varphi$ and $\lambda$ being the latitude and longitude in degrees, respectively. In our analysis, the derivative has been implemented with the central finite-difference method. In first numerical evaluations we tested different finite-differencing techniques (centered, forward, backward, and centralized with higher-order). We found that while forward and backward differencing is not recommendable, the central finite-difference method showed the smallest bias with respect to original wind, and was as a result chosen for the analysis. For details, please see Appendix A, Fig. A1.

**The geostrophic balance equation:** to derive wind fields outside the equator region, the geostrophic balance equation is used (e.g., Scherllin-Pirscher et al., 2014). The wind components are derived from geopotential $\Phi$, given on isobaric levels, resulting in the following formulation of the geostrophic zonal and meridional wind components, $u_g$ and $v_g$:

$$u_g \simeq -\frac{1}{f(\varphi)R_\mathrm{E}}\frac{\partial \Phi}{\partial \varphi}, \tag{3}$$

$$v_g \simeq \frac{1}{f(\varphi)R_\mathrm{E}\cos\varphi}\frac{\partial \Phi}{\partial \lambda}, \tag{4}$$

with $f(\varphi) = 2\Omega\sin\varphi$ being the Coriolis parameter, also implementing these derivatives with the central-difference method.

**Wind speed:** for both methods we calculated the wind speed as $V = \sqrt{u^2 + v^2}$, where the subscripts in our figures (Sect. 4 and Sect. 4.3) will indicate, whether the wind speed was derived from the equatorial balance ($eb$) or geostrophic ($g$) wind field approximation. Furthermore, the original wind speeds from the ERA5 reanalysis data have the subscript ($o$), indicating the original ERA5 wind data.

**Validation:** we derived the equatorial winds and the geostrophic winds for the complete globe. However, from our prior analysis we know, that between $\pm 5°$ latitude, the geostrophic approximation breaks down, since it is not the correct physical approximation for the wind retrieval (Nimac et al., 2023). In this region the equatorial-balance equation takes over. Hence, we indicate this latitudinal area in all our result figures with a light grey shaded area. Within this area, the validation of the equatorial balance equation is conducted, aiming to bridge the equatorial gap when deriving RO wind fields. The bias directly obtained from the equatorial balance equation is studied as the difference between ERA5 balanced ($eb$) and original ($o$) wind speeds, while the systematic difference is studied as the difference between RO and ERA5 balanced winds, as summarized in Table 1:

| Bias | Definition | Lat Range | Lon Range |
|---|---|---|---|
| Equatorial-balance bias | $\mathrm{ERA5}_{eb} - \mathrm{ERA5}_o$ | focus area $\pm 5°$ N/S | all |
| Systematic data bias | $\mathrm{RO}_{eb,g} - \mathrm{ERA5}_{eb,g}$ | ($eb$): focus area $\pm 5°$ N/S; ($g$): studied on complete globe | all |

**Table 1.** Definition of the equatorial-balance bias and systematic data bias, as well as our latitudinal and longitudinal range of focus in this specific study

.

The biases are validated for zonal wind ($u$), meridional wind ($v$), as well as wind speed ($V$). As introduced above, the target requirement for data quality in wind speed (and zonal wind component) is $\pm 2\,\mathrm{m\,s^{-1}}$, in line with WMO requirements WMO-OSCAR, 2023. The threshold is marked with dashed lines in the result figures.

## 3   Data sets

Monthly ERA5 reanalysis data (Hersbach et al., 2020) and monthly averaged RO OPSv5.6 data (Angerer et al., 2017; Steiner et al., 2020) from the year 2009 were used. This year was chosen for its high number of RO observations, representing a good approximation for later years, when the COSMIC-2 mission started (June 2019), which has an especially high number of observations in the tropics and the mid-latitudes. January 2009 was chosen as a representative month in the results section. All other months were analyzed as well and generally showed no major differences in behavior, which justifies the representative-month approach for most result discussions. As we also performed the analysis for the complete year 2009, for both ERA5 and RO data, we draw from these results to discuss aspects of seasonal and interhemispheric changes.

### 3.1   ERA5 reanalysis data

The ERA5 reanalysis data includes global 3D wind information and geopotential height, it is therefore the ideal data set to test the validity of the equatorial balance equation. It is available for a long time period and readily accessible via download from the Copernicus Climate Data Store (CCDS) (ECMWF-ERA5monthly). The data are available on 37 levels from $1000\,\mathrm{hPa}$ to $1\,\mathrm{hPa}$, on a $0.25°\mathrm{x}0.25°$ grid. Different grid resolutions were investigated for wind derivation to find the sensible spatial grid for the equatorial balance approximation. Fig. 1 shows the result for the zonal equatorial balanced wind, $u_{eb}$, where a) shows the result for the zonal wind component, tested for resolutions from $1.0°$ up to $5°$, and b) shows the difference to the original ERA5 zonal wind component, illustrated for January 2009.

The analysis in Fig. 1 illustrates that a grid spacing of $1°$ is counter productive, as the $u$ component shows large fluctuations. A grid of $2.5°$ or $3°$ results in similar values between derived and original wind fields. Furthermore, finer resolutions (temporal and spatial) increases the magnitude of the ageostrophic contributions, which are unbalanced (see, Bonavita, 2023). On the other hand, for a $5°$ spacing, the loss in resolution is noticeable.

As a result, we chose a $2.5°\mathrm{x}2.5°$ climatology for all further ERA5 wind investigations. The data sets with lower resolutions were derived from the original $0.25°\mathrm{x}0.25°$ grid, via cosine-weighted binning. The wind component data from the reanalysis is labeled $u_o$, $v_o$, and $V_o$, corresponding to the eastward-, northward wind component, and the wind speed. A line above the variable indicates a zonal average, e.g. $\overline{u}_o$. The wind components derived from the geopotential via the equatorial balance equation are referred to as $u_{eb}$, $v_{eb}$, and $V_{eb}$, or with a subscript $g$, when we used the geostrophic approximation.

### 3.2   Radio occultation data

In this study we focus on the potential of RO data to derive monthly meso-scale (2.5 x 2.5) wind products. A finer spatial resolution is, on the one hand, not recommendable for RO data and this time period. This would require more dense global

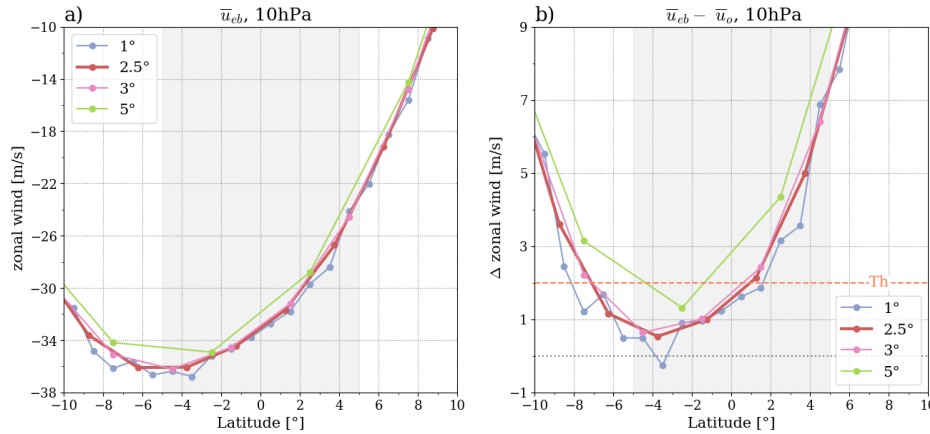

**Figure 1.** Influence of different spatial resolutions on the zonal-mean wind component. Panel a) shows the wind component, $u_{eb}$, panel b) the difference of the calculated wind to the ERA5 wind field zonal wind component $u_o$. The orange dashed line marks the $2\,\mathrm{m\,s^{-1}}$ threshold and the results are shown for January 2009.

coverage with daily RO events, which is not available up to now (see also Angerer et al. (2017); Ladstädter et al. (2023)). On the other hand, as a further physical reason, the geostrophic and equatorial balance will also not hold well at higher temporal or spatial resolution, leading to larger ageostrophic contributions. We analyze the monthly RO climatologies data from multi-satellite missions in the year 2009. The RO phase data were derived at UCAR/CDAAC (University Corporation for Atmospheric Research/COSMIC Data Analysis and Archive Center), while the further processing to geopotential height,

$Z(p)$, calculated on isobaric surfaces $p$, was performed using the WEGC Occultation Processing System OPSv5.6 (Angerer et al., 2017; Steiner et al., 2020). The WEGC OPSv5.6 retrieval system processes the atmospheric parameters as a function of altitude or geopotential height, based on the refractivity equation, the equation of state, and the downward integration of the hydrostatic equation. The physical atmospheric parameters (e.g., physical pressure) are derived using a moist-air retrieval algorithm, which combines the individual profiles with background information by optimal estimation; see Li et al. (2019)

for details. The conversion to geopotential $\Phi(p)$ is defined as $\Phi(p) = Z(p) \cdot g_0$, where $g_0 = \pm 9.806\,65\,\mathrm{m/s^2}$, being the global standard gravity at mean sea level. In the year 2009, data is available from the following missions: Satélite de Aplicaciones Científicas (SAC-C) (e.g., Hajj et al., 2004), Gravity Recovery And Climate Experiment (GRACE-A) (e.g., Beyerle et al., 2005), Formosa Satellite Mission 3/Constellation Observing System for Meteorology, Ionosphere, and Climate (Formosat-3/COSMIC) (e.g., Anthes et al., 2008), and from the Meteorological Operational Satellite (MetOp-A) (e.g., Luntama et al.,

2008). The year 2009 was chosen as a representative data set to analyze the wind dynamics within a full year, having at the same time the advantage of a rather high occultation statistics, due to the fully available six-satellite constellation of the Formosat-3/COSMIC mission (Angerer et al., 2017).

     The monthly climatologies were produced on a 2.5° x 2.5° grid, using a $600\,\mathrm{km}$ radius which corresponds to the distance from the grid point, defined as the center location of the area of influence, within which the profiles contribute to the grid

point mean. In performing the averaging, the profiles are weighted according to their distance from this center location with a bivariate (latitude-longitude) gaussian function which peaks at the center and features a standard deviation of 150 km along latitude and 300 km along longitude, respectively. Details are given in the presentation by Ladstädter (2022). The geopotential climatologies $\Phi(p)$ are available from $1000\,\text{hPa}$ to $5\,\text{hPa}$, on 147 levels. The geopotential was further binned to a $5° \times 5°$ grid, using a cosine weighted binning. From this larger bins, the equatorial balanced winds were calculated. Tests revealed that a Gaussian smoothing with a $5°$ longitudinal smoothing window improved the wind data estimation and the systematic difference decreased. This smoothing was therefore applied to the equatorial-balance wind fields derived from RO data. The larger binning was performed to avoid small fluctuations in the wind data, which required larger climatologies. Regarding geostrophic winds, the $2.5° \times 2.5°$ grid could be maintained. The most prominent difference in the computation between equatorial and geostrophic winds is that the former requires a double derivative, while the latter requires a single derivative. Hence, small fluctuations in the data are enhanced for the equatorial balance equation, which makes the derivation of winds a bigger challenge. However, we emphasize at this point, that due to the COSMIC-2 mission (start in June 2019), which provides a higher sampling in the tropics, the potential of finer resolutions is given (Schreiner et al., 2020; Ho et al., 2020).

For the comparison between calculated ERA5 and RO wind, an interpolated ERA5 reanalysis data set with 364 levels from $1000\,\text{hPa}$ to $10\,\text{hPa}$ was used. Since RO data were binned to a $5°\text{x}5°$ grid (see Sect. 3.2, to have a sufficient number of observations per grid cell), the ERA5 data set was also transferred to a $5°\text{x}5°$ grid, using cosine weighted binning. For this specific data set the prefix ERA is used.

## 4 Results and discussion

To validate the equatorial balance equation, the zonal and meridional wind components, and wind speed were calculated according to the equations introduced in Sect. 2. We analyze the bias from the equatorial balance equation in Sect. 4.1. The systematic bias between the observation-based RO data set and the reanalysis data set, is investigated in Sect. 4.2. Hereby, the potential of RO wind products over the equatorial region is tested. The results on closing the gap across the equator are discussed in Sect. 4.3. Furthermore, all vertically-resolved plots are shown down to 800 hPa, since our focus is the free atmosphere, excluding the atmospheric boundary layer and hence frictional force.

### 4.1 ERA5 wind validation

To test the quality of the equatorial balance equation, both wind components individually, and the total wind speed are compared to the original wind field in ERA5. Fig. 2 shows the original wind and the wind component/wind speed calculated with the equatorial balance equation, as well as the respective difference for January 2009. The analysis was performed in $20°$ meridional bands. We find that the zonal wind component shows maximum magnitudes larger than $-30\,\text{m\,s}^{-1}$ around $8\,\text{hPa}$, and up to $10\,\text{m\,s}^{-1}$ between $50\,\text{hPa}$ to $30\,\text{hPa}$ for both, original and derived, zonal wind (Fig. 2a and 2b).

The analysis of the difference between computed and original ERA5 fields illustrates generally a good agreement within $\pm 2\,\text{m\,s}^{-1}$ in the stratosphere, reaching $\pm 5\,\text{m\,s}^{-1}$ when the absolute magnitudes reach maximum values; i.e., around $8\,\text{hPa}$

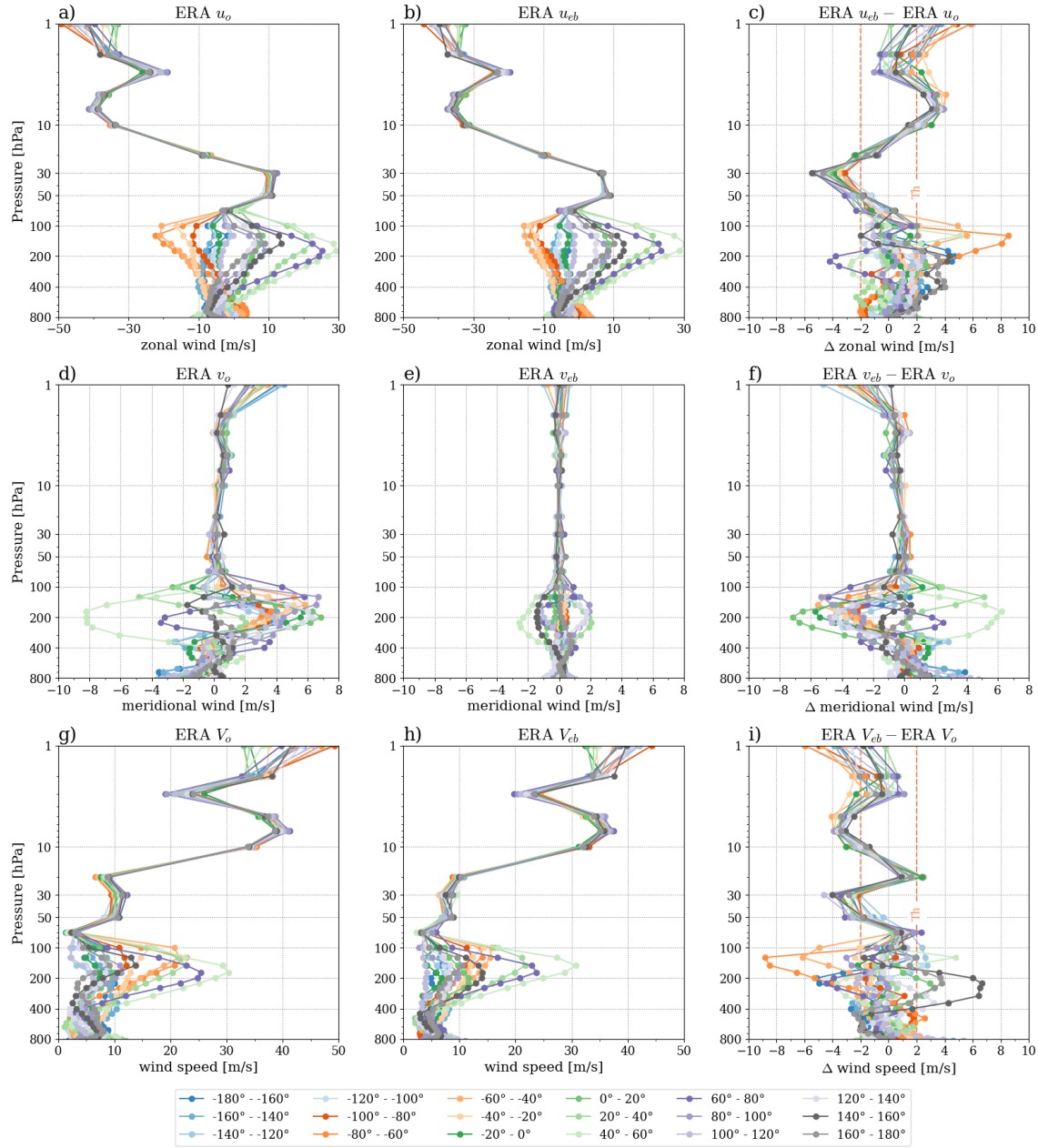

**Figure 2.** Panels a), d) and g) show the $u$, $v$ and $V$ component of the original ERA5 data (first column). Panels b), e) and h) show the wind components calculated with the equatorial balance approximation (second column). The last column illustrates the difference between the derived and the original wind data from ERA5. The wind components and wind speed are studied for $20°$ meridional bands and a $\pm 5°$ latitudinal averaging. The data are from January 2009.

and between $50\,\mathrm{hPa}$ to $30\,\mathrm{hPa}$, respectively. Furthermore, the analysis shows that the different longitude bands coincide in the stratosphere. Also in the middle to upper troposphere, the difference between the derived winds and the original winds is predominantly within the threshold, however, the individual longitude bands do not coincide anymore (pressures higher than the $100\,\mathrm{hPa}$ level).

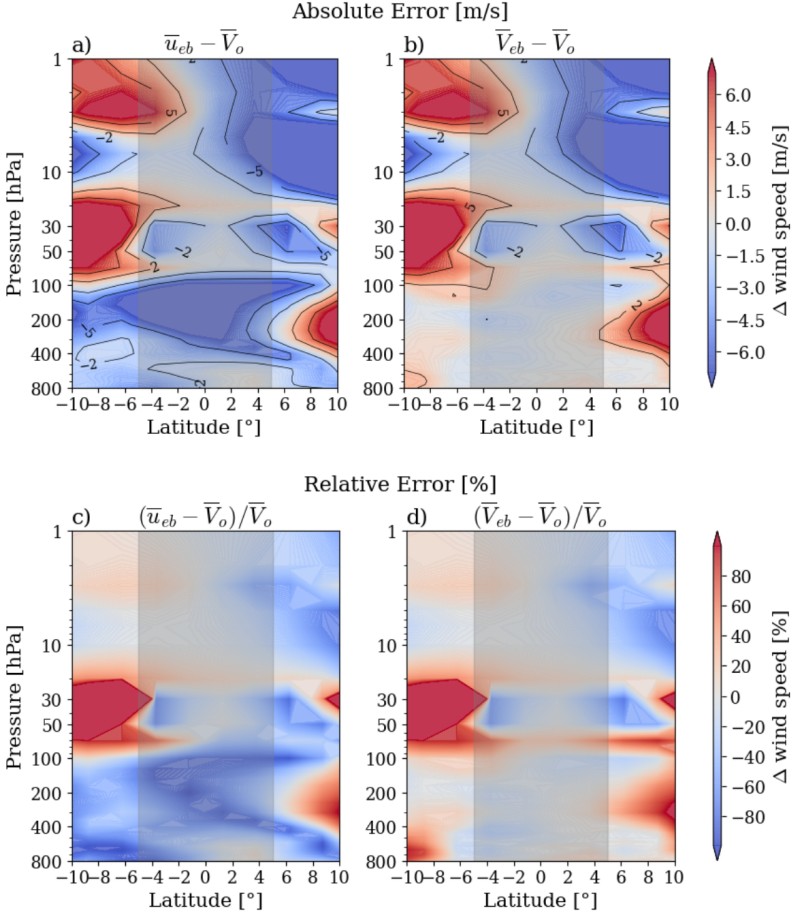

**Figure 3.** Absolute (first row) and relative difference (second row) between derived and original ERA5 zonal-mean total wind speed, shown as a vertical-latitudinal cross sections. The first column uses only the zonal-mean component as an approximation for wind speed, while the second column includes the zonal and meridional component to estimate the total wind speed. The data are from January 2009.

In Fig. 2, middle row, we show the meridional wind component, where the magnitudes of the wind speed are much smaller. We show here the results of the meridional wind component for all longitude bands. In further analysis we will only present results based on one exemplary longitude band around the prime (Greenwich) meridian (-10° to 10° longitude), keeping notice that we had studied the other sectors as well, which qualitatively showed similar behavior. First of all the meridional wind is very small in the tropical stratosphere (see Fig. 2d and 2e). Second, the meridional wind is much smaller than the zonal wind

(close to zero compared to the zonal wind). Third, even the bias in the zonal component is larger than the meridional component itself, and finally, since the meridional wind is very small it cannot be well represented by the equatorial balance equation. In the troposphere, the meridional wind speed increases to values around $\pm 4\,\mathrm{m\,s^{-1}}$. Analyzing the equatorial-balance bias shows that the difference fluctuates with amplitudes of about $\pm 2$ to $4\,\mathrm{m\,s^{-1}}$ in the tropical troposphere (Figure 2f).

Finally, we study the total wind speed ($V_{eb}$, Fig. 2 bottom row). To this end the question is, if the meridional wind component has an added value for the total wind speed ($V = \sqrt{u^2 + v^2}$), since its magnitude is close to zero in the stratosphere. In this first analysis we included the meridional wind component in the computation of wind speed, finding that it was possible to derive the wind fields close to the original wind speed (Fig. 2g and 2h), and within our defined threshold (Fig. 2h), from the middle troposphere up to the stratosphere.

$$\mathrm{ERA}\ \overline{V}_{eb} - \mathrm{ERA}\ \overline{V}_o$$

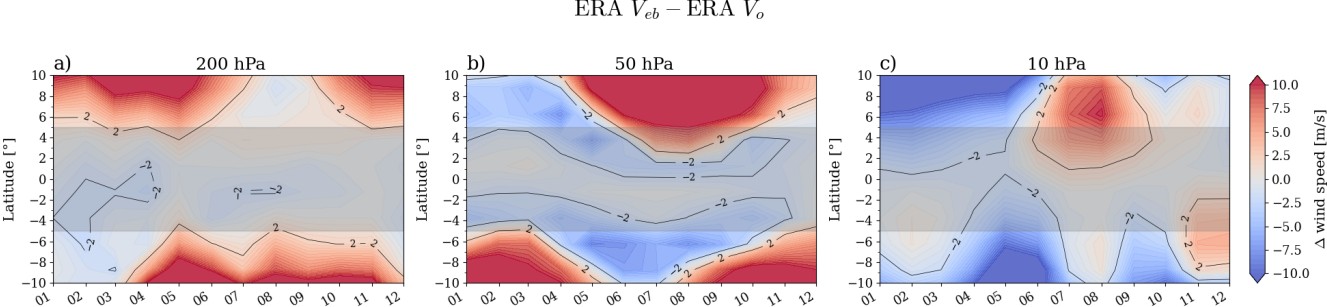

**Figure 4.** Seasonal development of the zonal-mean total wind speed bias resulting from the equatorial balance equation for the year 2009, studied for ERA5 data on three representative pressure levels.

To better understand the potential added value of the meridional wind component in total wind speed, we study in Fig. 3 the impact of the $v$ component on the final product in more detail. We show a vertical-latitudinal cross section and approximate the zonal-mean wind speed by only using the zonal wind component (first column), compared to including both components for the zonal-mean wind speed estimate (second column). We study the absolute (top row) and relative (bottom row) difference to the original ERA5 data. In the absolute difference, the two estimates of zonal-mean wind speed show very similar results in

the stratosphere. This is because the meridional component is close to zero in magnitude, having only a negligible impact on the total wind speed.

The situation changes in the troposphere at pressures higher than the $100\,\mathrm{hPa}$ level. The zonal-mean wind speed clearly improves when including the meridional wind component for the estimate of wind speed (cf. Fig. 3a to Fig. 3b). The differences between derived and calculated wind speed are mainly within the target threshold of $\pm 2\,\mathrm{m\,s^{-1}}$ when including the

meridional wind. Also the relative difference (bottom row) illustrates this clear improvement in the zonal-mean wind speed data, when comparing Fig. 3c to Fig. 3d. This result indicates that while the meridional component itself is not well estimated, the calculation of the zonal-mean total wind speed shows a somewhat closer agreement from including also this small wind component in the troposphere. That is, the slight underestimation left by the zonal wind speed in the troposphere (see Fig. 3a,c) is mitigated by the inclusion of the meridional wind speed, which brings the approximated zonal-mean total wind speed closer

to the original one (see Fig. 3b,d). The reason for this small co-benefit to the zonal-mean total wind speed is that the averaged meridional wind speed is estimated by the balance approximation at about the right (small) magnitude in the troposphere. However, in the stratosphere, the close-to-zero meridional wind brings in no further improvement. Nevertheless, we caution that, in addition to finding the meridional wind estimates not viable to benefit also longitude-resolved wind speeds, we also find them not viable to reconstruct wind direction (i.e., wind vectors on top of wind speed estimates). The reason is that the direction of the approximated meridional wind at individual grid points is nearly as often found opposite as it is found aligned with the original wind direction.

Furthermore, we investigate the bias resulting from the equatorial balance approximation for the complete year 2009 (Fig. 4). We show these results on the three representative levels, $200\,\mathrm{hPa}$, $50\,\mathrm{hPa}$ and $10\,\mathrm{hPa}$, representing the tropical upper troposphere, lower stratosphere and middle stratosphere, respectively. Across all seasons, the equatorial balance approximation shows best results in the absolute difference at lower altitudes. For the lower stratosphere, the region below the bias threshold shifts away from the equator with the seasons, with an offset in the direction of the winter hemisphere. For $10\,\mathrm{hPa}$, the middle stratosphere, the approximation is least accurate during northern hemisphere summer months.

## 4.2 RO wind validation

In this section we investigate the systematic data bias between RO and ERA5 derived wind fields. To remind the reader, we use a two-step approach to assess the potential of using the equatorial-balance equation for RO wind field derivation across the equator (see also Table 1). In a first step, we decomposed the analysis into the bias originating from the approximation itself (first step, only ERA5, Section 4.1). In a second step, we now assess the systematic bias between the two data sets (ERA5 and RO), to understand where differences between them enter. First of all, we observe that for RO $u_{eb}$, RO $v_{eb}$ and RO $V_{eb}$ the spatial patterns look very similar to the wind fields calculated with ERA5 data, see Fig. 5 top to bottom row, respectively. Between 30 hPa and 10 hPa the bias between the two data sets and for the zonal wind ($u_{eb}$, top row) lies between $\pm 2\,\mathrm{m\,s^{-1}}$ and $\pm 5\,\mathrm{m\,s^{-1}}$, increasing towards higher altitudes (Figure 5c). A possible reason could be, that the impact of the residual ionospheric error, as well as measurement noise increase towards higher altitudes for RO data (e.g., Danzer et al., 2013, 2018; Liu et al., 2020, 2024). With the exception of this region and the boundary layer, the target threshold of $\pm 2\,\mathrm{m\,s^{-1}}$ is rarely exceeded, and the wind speed differences are very small between the two data sets. When considering the differences between the two data sets for meridional wind and wind speed (Figure 5f and 5i) it is seen that these also generally reside within $\pm 2\,\mathrm{m\,s^{-1}}$. Nevertheless, due to the already small absolute magnitudes of the meridional wind (Figure 5d and 5e) it is also clear that this component itself is not well reproduced. However, the zonal-mean total wind speed (bottom row) still shows an improvement in the troposphere from including both wind components (Figure 5i), with the dominant contribution coming from the zonal component

Nimac et al. (2023) found that the bias between RO $V_g$ and ERA $V_g$ decreased after 2016, when ERA5 undertook a major observing system change. It is reasonable to assume that similar behavior could be observed for RO $V_{eb}$ and ERA $V_{eb}$. As the number of RO satellite missions in operation changes, there are fluctuations in the number of available RO profiles to aggregate for a given time period. The years 2008 and 2009 show a high number of daily occultations (roughly 2500 to 3000 events).

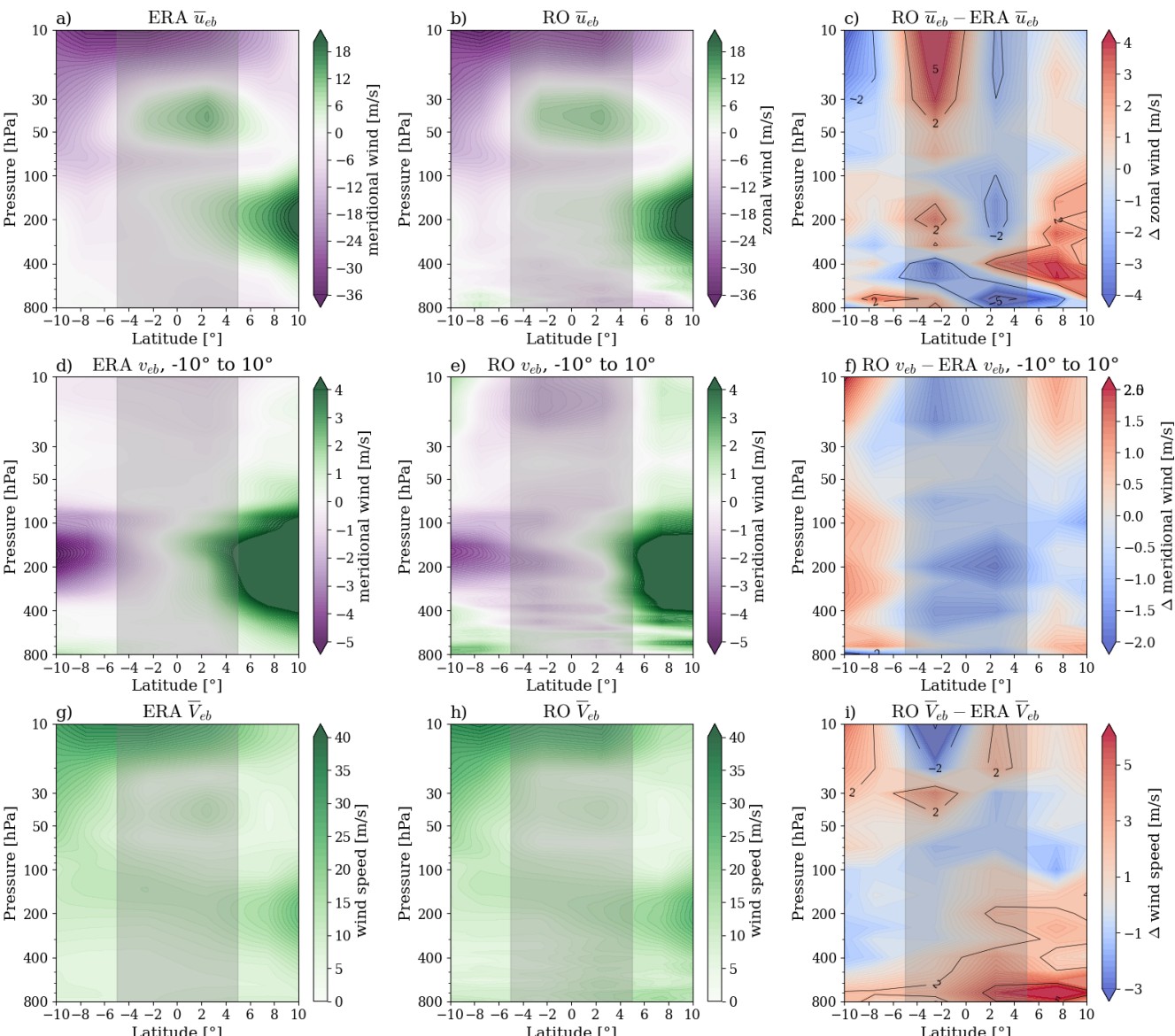

**Figure 5.** Panels a), d) and g) show the $u$, $v$ and $V$ components calculated with the equatorial balance approximation for ERA5 data, while panels b), e) and h) show the the same using RO data. The bottom row illustrates the difference between the values calculated between RO data and ERA5 data. Note that $u$ and $V$ are plotted as a zonal average, while the $v$ component is shown exemplary for the longitude sector $-10°$ to $10°$. The data are from January 2009.

In the years 2011 and 2012 there is a significant drop in the daily available occultations (rougly 1500 events), see Angerer
et al. (2017). In those years we have no further data from the F3C-FM3 satellite, and also the SAC-C mission ended. However, with the launch of the COSMIC-2 mission in 2019, which is specifically designed for a high coverage in the tropics up to the

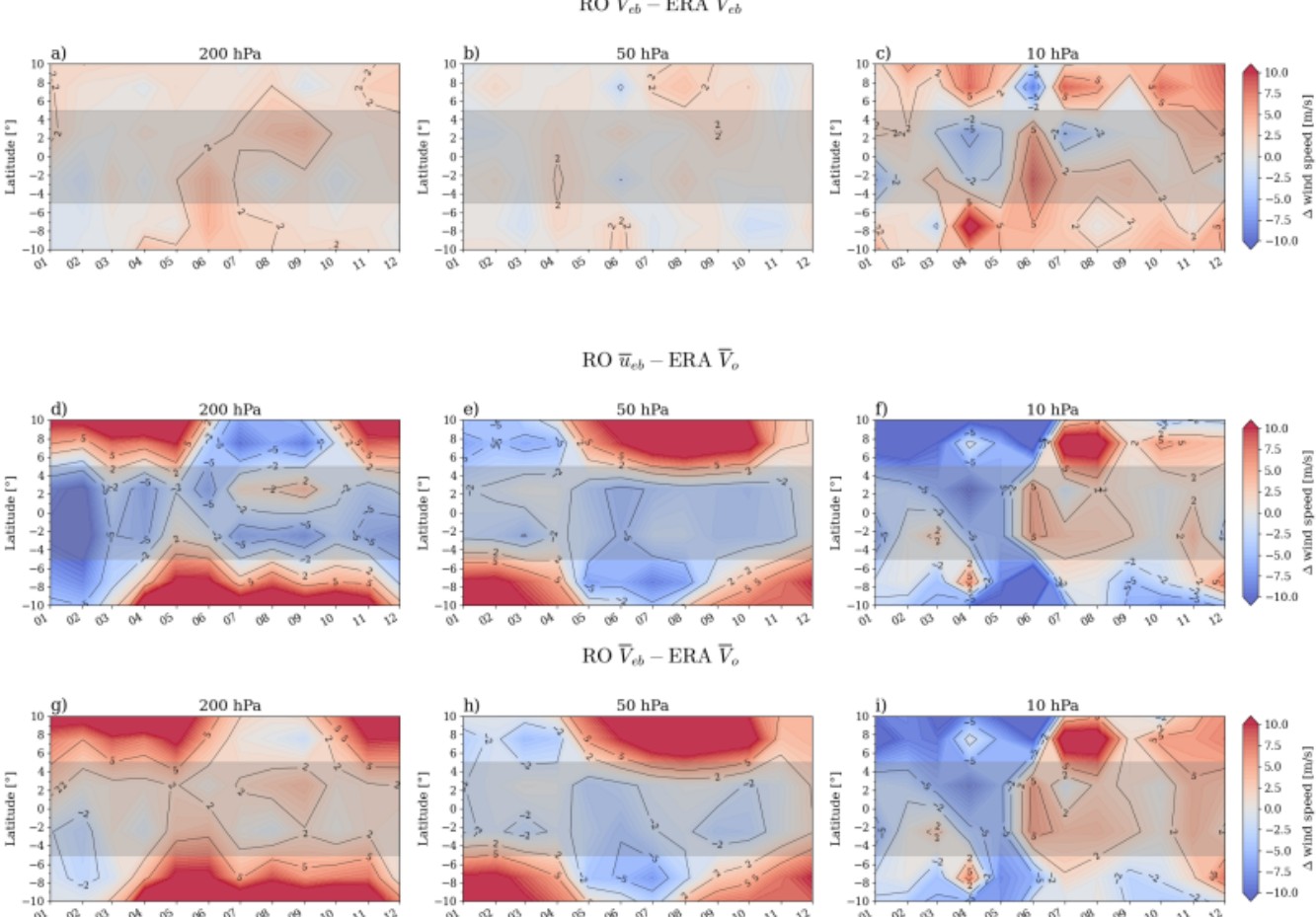

**Figure 6.** Seasonal development of the systematic data bias (first row) between RO and ERA5 data, studied for the year 2009 on three representative pressure levels. Second and third row illustrate the seasonal development of the zonal-mean total wind speed bias, using for the former only the zonal component as an approximation for wind speed, while the latter includes the zonal and meridional component in the estimate.

mid-latitudes (Schreiner et al., 2020; Ho et al., 2020), the accuracy of RO data in the equatorial region will further increase. This possibly also allows to use a 2.5° x 2.5° wind field grid in future studies.

In a final analysis we investigate the seasonal development of the systematic (first row) and total wind speed bias (second and third row) for the complete year 2009 (see Fig. 6). With respect to the systematic data bias (first row) between RO (RO $V_{eb}$) and ERA5 (ERA $V_{eb}$) there is little to no deviation from the upper troposphere (200 hPa) to the lower stratosphere (50 hPa) all year. Only at the 10 hPa level we observe somewhat larger deviations, most notable in the northern hemisphere summer months. The numbers of RO profiles accumulated to generate the monthly RO data set dropped by around 33% in June 2009 compared to other months in the same year. This possibly decreases the data quality and therefore we observe an increase in

the systematic data bias (e.g., Scherllin-Pirscher et al., 2011; Schwarz et al., 2017). The bias is within $\pm 2 \,\mathrm{m\,s^{-1}}$ and $\pm 5 \,\mathrm{m\,s^{-1}}$, indicating that towards high altitudes the wind speed retrieval over the tropics gets more challenging. We still see a potential for improvements in the ongoing work, by correcting residual biases of RO data in the upper stratosphere (Danzer et al., 2021; Liu et al., 2020).

The second and third row of Fig. 6 examines the difference between RO computed winds relative to the original ERA5 winds.
The second row only uses the zonal wind component for the estimate of the zonal-mean total wind speed, while in third row both the zonal and meridional components are included. When comparing the two rows, the results illustrate an improvement for zonal-mean wind speed when including the meridional wind (third row). The geographical band we are focusing on is between $\pm 5°$ latitude (light shaded grey area). Within this area the bias clearly decreases between RO and ERA5 wind speed, at the $200 \,\mathrm{hPa}$ and $50 \,\mathrm{hPa}$ levels. At the $10 \,\mathrm{hPa}$ level the pattern is similar between the second and third row, illustrating the
decreasing influence of the meridional wind component, see also the discussion in the previous Sect. 4.1.

Summarizing the results of the current and previous section, meso-scale climate wind field derivation was possible across the equator using RO data, when focusing on its core vertical region of high quality and resolution. Furthermore, we found that while the meridional component itself is not well estimated, it is the zonal-mean total wind speed in the troposphere that shows a closer agreement with the corresponding original wind speed when including both components, while in the stratosphere the
325 meridional component's influence becomes negligible.

## 4.3 Closing the equatorial gap

In this final results section, we aim to bridge the wind field gap over the equator to complete with a wind field product over the complete globe. For this reason, we have once more a closer look at the zonal and meridional wind, as well as wind speed, at the three respective pressure levels 10 hPa, 50 hPa, and 200 hPa (first to third row, Fig. 7). In Fig. 7 we compare the computed
winds, i.e., equatorial balance ($eb$) and geostrophic balance ($g$) RO and ERA5 winds, to original ($o$) ERA5 winds (black solid line). We analyze how the equatorial balance and geostrophic balance approximations bridge over the equator, thereby finding some interesting results. We observe that the zonal geostropic wind ($\overline{u}_g$) actually does not break down between $\pm 5°$, neither for RO or ERA5 computed winds. The results for $\overline{u}_g$ are actually closer to the original wind (black line, Fig. 7a, d, and g), than the computed zonal equatorial balanced winds ($u_{eb}$, RO and ERA5). The component primarily responsible for the increase of
the geostrophic bias over the equator is the meridional wind component ($v_g$), showing the largest differences with respect to the original ERA5 meridional component ($v_o$) at 10 hPa, decreasing towards 200 hPa (Fig. 7b, e, and h). Here, the equatorial balance solution ($v_{eb}$, RO and ERA5) clearly better reproduces the ERA5 meridional winds, having the smallest bias at 200 hPa, with an increasing bias towards 10 hPa. Since the geostrophic meridional wind drives the equatorial breakdown ($v_g$), as a result, also the geostrophic wind speed ($V_g$) shows larger biases over the equator, while the equatorial wind speed ($V_{eb}$) is a
better fit between $\pm 5°$ (Fig. 7c, f, and i).

Finally, we use our knowledge from the prior analysis to compute a complete global wind field data set, using RO data. In Fig. 8 we show the result based on the four seasonal representative months; January, April, July, and October. The wind fields are illustrated as a vertical cross section from 800 hPa to 10 hPa, dependent on latitude. The l.h.s. of this plot (Fig. 8a, b, c,

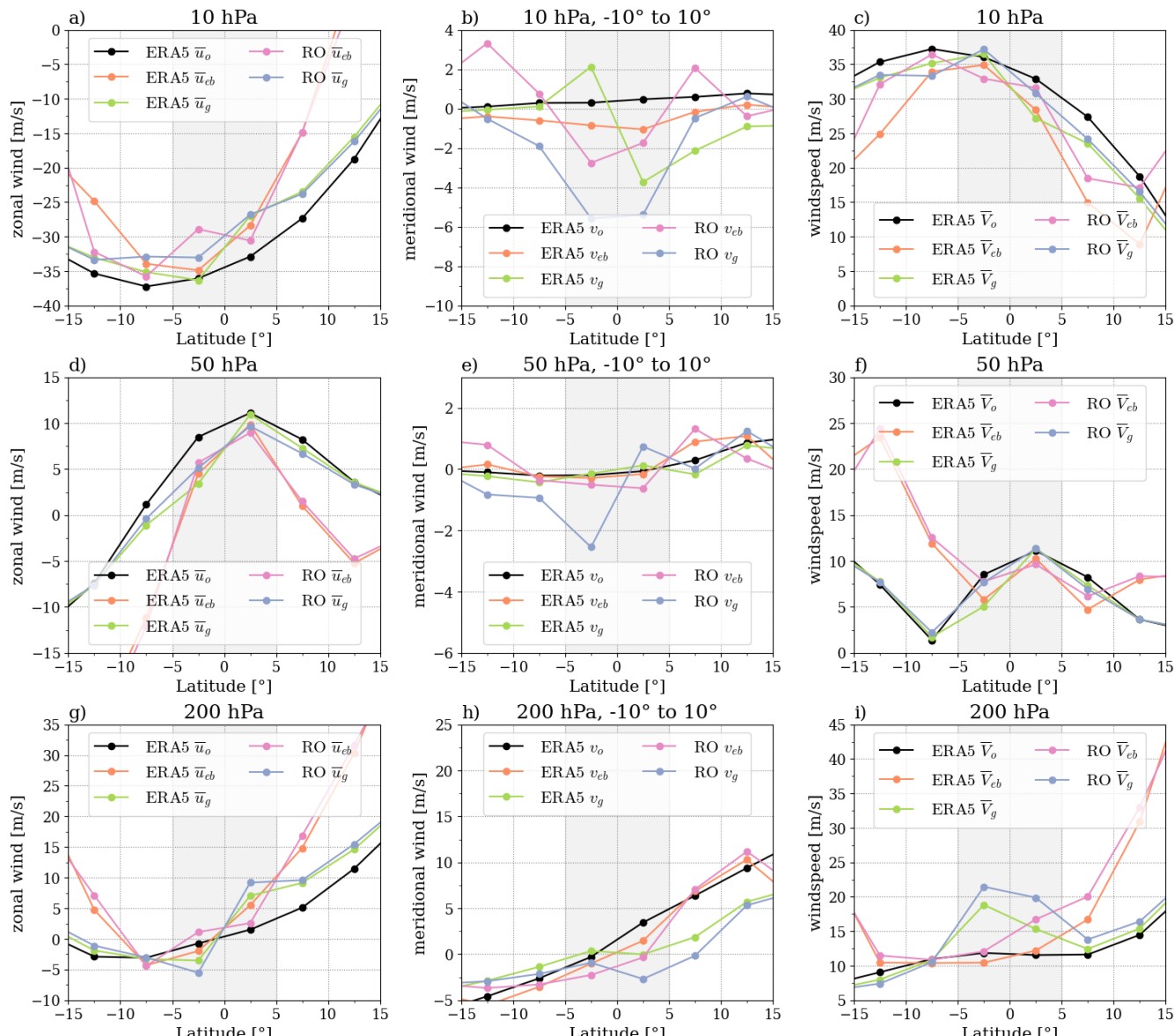

**Figure 7.** Detailed analysis of zonal wind ($u$, first column), meridional wind ($v$, second column), and wind speed ($V$, third column), at the three pressure levels 10 hPa, 50 hPa, and 200 hPa (first to third row). Results are shown for the original ($o$), equatorial-balanced ($eb$), geostrophic ($g$) ERA5 data and RO data ($eb$, $g$). $u$ and $V$ are plotted as the zonal mean, while the $v$ component is calculated as a mean from the longitudinal sector $-10°$ to $10°$. The data are from January 2009.

d) shows the bias between the computed RO wind fields, relative to ERA5 computed wind fields. Between $\pm 5°$ we use the equatorial balance equation for the calculation of the wind speed ($V_{eb}$), while outside this latitude band the geostrophic balance approximation is applied ($V_g$). We find that the bias between the two data sets is very low, with differences dominantly less

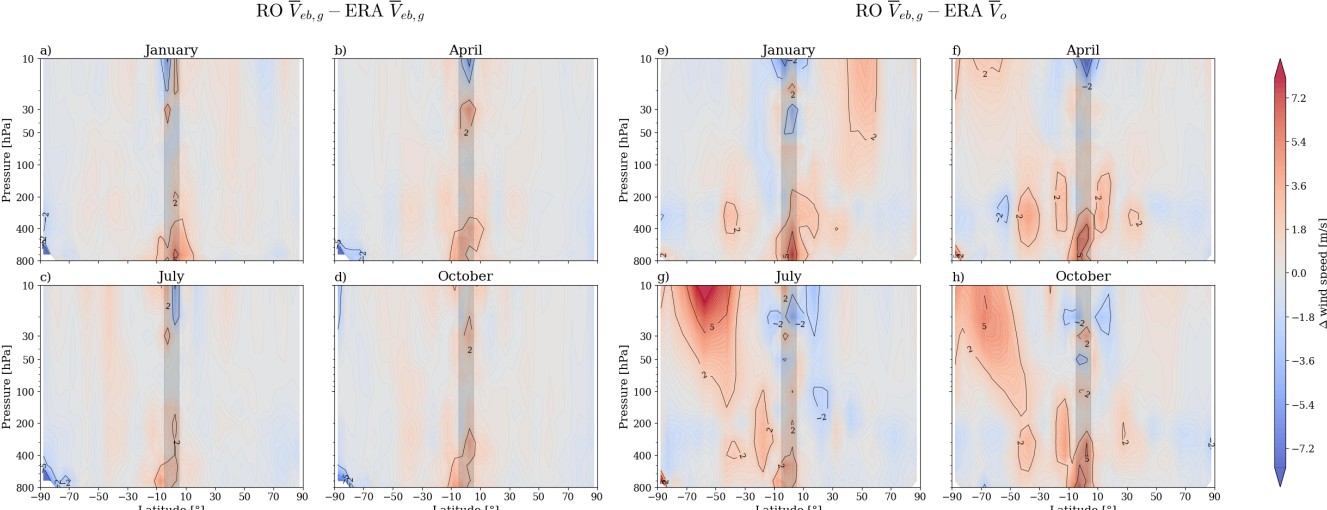

**Figure 8.** Zonal-mean systematic data bias, panels a) to d), and zonal-mean wind speed bias, panels e) to h). To construct the RO and ERA wind fields, the equatorial balance equation is used inside the equatorial band (|Lat.| < 5, while outside this region the geostrophic balance approximation is used. The data are from January 2009.

than $\pm 2\,\mathrm{m\,s^{-1}}$. In the equatorial latitude band we find small exceedances in the lower troposphere, while the upper troposphere and lower stratosphere (UTLS ) are very close to ERA5. This feature clearly relates to the core region of high-quality RO data, which is in the upper troposphere and lower stratosphere. With decreasing altitude and therefore increasing moisture
content, the retrieval of atmospheric parameters relies increasingly on background information (e.g., Li et al., 2019). The RO information dominates between about 8 km to 35 km in the tropics (e.g., Scherllin-Pirscher et al., 2011).

As a final comparison, we show on the r.h.s. of Fig. 8e, f, g, h, the respective RO wind fields relative to the original ERA5 wind data. We can conclude that the quality of the wind fields is especially above 500 hPa very good, and mostly within the target of $\pm 2\,\mathrm{m\,s^{-1}}$. Outside this latitude band we apply the geostrophic approximation, and find also a high wind speed quality.
Exceptions are the stratospheric polar jet stream and the sub-tropical jet stream, where larger deviations are found. However, this was not part of this specific analysis. More information can be found in Nimac et al. (2023, 2024).

## 5   Summary and Conclusions

In this study we investigated the potential of radio occultation (RO) data for climate-oriented wind field monitoring in the tropics, with a specific focus on the equatorial band within $\pm 5°$ latitude. We analyzed the equatorial balance equation within
this band and computed RO wind fields at a 5° x 5° resolution. In a wider range over the tropics, we computed the RO wind fields, using the geostrophic approximation, on a higher-resolved 2.5° x 2.5° grid. We also calculated the winds using ERA5 data, applying the same physical equations and resolutions for the comparison analysis (basic ERA5 resolution 2.5° x 2.5°).

In a first step, we analyzed the bias solely resulting from the equatorial balance approximation, by studying the difference between computed winds from ERA5 geopotential and original ERA5 winds. In a second step, we compared the balance-derived approximate RO winds to the approximative ERA5 winds. This two-step approach allowed to separately study the bias resulting from the approximation itself, and the systematic bias between the two data sets. We also analyzed how the geostrophic and equatorial-balanced zonal winds, meridional winds, and wind speeds bridge across the equator, to understand which wind component drives the geostrophic breakdown over the equator.

The results showed that we could successfully apply the equatorial balance equation for the RO wind field computation across the equator. For the $u$ component this was already examined by Healy et al. (2020) in a zonal analysis. In our study we were able to resolve the zonal wind by a 5° x 5° latitude x longitude grid. Furthermore, we included in this analysis the meridional wind component, as well as total wind speed, applying the same grid resolution. However, the meridional wind component was challenging, since its wind speed is in general an order of magnitude smaller than the zonal wind in the troposphere, while in the stratosphere its contribution is negligible. Hence, a wind flow with small magnitudes and also changes in the direction of the flow (changing sign) is challenging to reproduce. We find in this respect the equatorial balance approximation not suitable for the reconstruction of wind directions and of longitude-resolved wind speeds in the troposphere.

Nevertheless, the analysis clearly showed that calculating both, the zonal and meridional wind components, resulted in a higher quality of the zonal-mean total wind speed in the troposphere. In the stratosphere, total wind speed is governed by the zonal component and no added value furnished by the meridional component. In general, the equatorial balance approximation works best in the stratosphere. The biases were mostly within the target requirement of $\pm 2\,\mathrm{m\,s^{-1}}$, decreasing a bit in quality upward towards the 10 hPa level. In this respect we emphasize that the COSMIC-2 mission (Schreiner et al., 2020; Ho et al., 2020), with dense RO event coverage in the tropics, will substantially improve RO wind fields in this area. Furthermore, the potential exists in this case to refine to the desirable 2.5° x 2.5° resolution for the wind field data.

Another important result was found when analyzing the individual wind components ($u$ and $v$) and total wind speed ($V$), for geostrophic ($g$), equatorial-balanced ($eb$), and original ($o$) winds, comparing RO and ERA5 data. We found that the dominant wind component, which drives the geostrophic breakdown, is the meridional wind, while the zonal geostrophic wind works well also across the equator. The geostrophic zonal wind ($u_g$) performed even a bit better in quality than the equatorial zonal wind ($u_{eb}$). Nevertheless, the equatorial balance approximation works as a robust solution of the wind equation just within the equatorial $\pm 5°$ band. Outside this band, it is no longer a valid approximation and hence breaks down. We tested also combinations of the total wind speed in this region, as a vector sum of zonal geostrophic wind and meridional equatorial wind in a specific altitude range. The results were quite satisfactory as well (not shown), but to explore in detail a most suitable combined wind field construction needs to be part of a future study.

To summarize, we found encouraging results in that we revealed that RO data do indeed have good potential for long-term wind field monitoring over the complete globe, including across the equator. A meso-scale climate resolution (5° x 5° latitude x longitude grid) was possible to be demonstrated for the RO data in this specific region for wind speed, with evidence for added value from their accuracy and high resolution, as well as their long-term stability.

*Data availability.* All computed wind field data for the year 2009 can be found under WEGC-cloud. The folder contains the following three files: (i) the wind field calculated from the WEGC Occultation Processing System OPSv5.6 RO data, (ii) the wind field calculated from ERA5 reanalysis data and further interpolated at the WEGC, and (iii) the wind field calculated from the download from the Copernicus Data store. The original RO OPSv5.6 data are available under WEGC-OPSv5.6

**Appendix A**

We tested different finite-differencing techniques (centered, forward, backward, and centralized with higher-order). We found that while forward and backward differencing is not recommendable (for truncation errors being of order $\mathcal{O}(h)$), centralized and higher-order centralized methods show very similar results when using ERA5 data on a 2.5° x 2.5° grid.

The standard central (Eq. A1) and higher-order central (Eq. A2) finite difference methods for the second-derivative (curvature) operator $\frac{\partial^2}{\partial x^2}$, on a function $f(x)$, with $h$ being the step size of the numerical grid, were used in our study through the following conventional formulations:

$$\frac{\partial^2 f(x)}{\partial x^2} \simeq \frac{f(x+h) - 2f(x) + f(x-h)}{h^2} + \mathcal{O}(h^2) \,, \tag{A1}$$

$$\frac{\partial^2 f(x)}{\partial x^2} \simeq \frac{-f(x+2h) + 16f(x+h) - 30f(x) + 16f(x-h) - f(x-2h)}{12h^2} + \mathcal{O}(h^4) \,. \tag{A2}$$

Fig. A1 illustrates the bias differences that result between these two finite-difference methods. The left column shows the bias based on ERA5 data (balance-derived versus original) while the right column shows the impact of the bias based on RO data (balance-derived versus original). We inspect the two relevant bias types, (i) a zonal mean ($\overline{V_{EB}} - \overline{V_O}$) between derived wind field and original ERA5 wind, and (ii) the local bias within single grid cells, taking the zonal mean afterwards ($\overline{|V_{EB} - V_O|}$).

We find for the ERA 5 data (left column), computed on a 2.5° x 2.5° grid, that the local approximation bias at individual grid points ($\overline{|V_{EB} - V_O|}$) is slightly smaller when using the standard central method, while the zonal mean bias improves a bit with the higher-order method. These biases are amplified when using the RO data available on a 5° x 5° grid (right column). Here the difference in the local bias is found larger, with the standard central method outperforming the higher-order method. This larger local bias of the higher-order 5-point method (Eq. A2) compared to the standard 3-point method (Eq. A1) is likely caused by the fairly large latitudinal range of the former across the central grid point, spanning across four 5° steps.

For the zonal-mean bias, again the higher-order method performs somewhat better, with the quality depending on altitude level and month. Overall, since the equatorial balance approximation is, strictly speaking, only fully valid at the equator, the approximation error from including data points outside of the $\pm 5°$ equator band is considered larger than the gain from applying the higher-order method. For this reason, the standard centered differencing method was finally chosen as the primary method for the respective data analyses in this study.

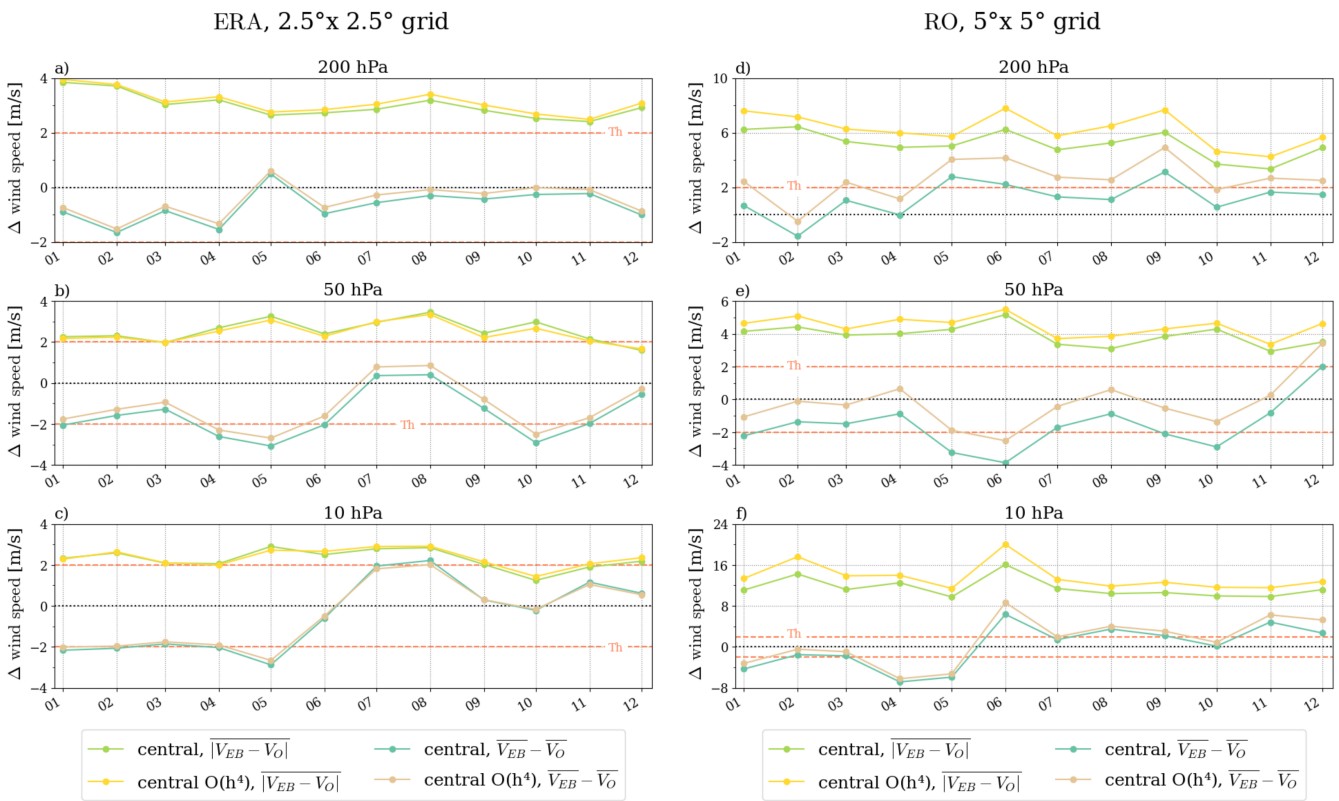

**Figure A1.** Figure illustrating the bias resulting from using two different finite difference methods, i.e., comparing standard centralized and higher-order centralized differencing. To show different relevant aspects of averaging, the bias is computed as the zonal-mean bias between balance-derived RO or ERA5 wind field and original ERA5 wind field ($\overline{V_{EB}} - \overline{V_O}$), and also as the local bias within individual grid cells, taking afterwards the zonal mean ($\overline{|V_{EB} - V_O|}$). Panels a, b and c show the bias for ERA5 data, while d, e, and f show the bias for RO data.

*Author contributions.* Conceptualization: JD, GK; Data curation: MP; Formal analysis: MP, JD; Funding acquisition: JD; Methodology: JD, GK; Supervision: JD, GK; Validation & Visualization: MP, JD, GK; Writing – original draft preparation: JD, MP; Writing – review & editing: JD, GK

*Competing interests.* The authors declare that they have no competing financial or personal interests.

*Acknowledgements.* We thank the UCAR/CDAAC RO team for providing RO excess phase and orbit data and the WEGC RO team for providing the OPSv5.6 retrieved profile data. We particularly thank F. Ladstädter (WEGC) for providing the monthly gridded climatology data, and I. Nimac for supplying the initial scripts, from which the wind field derivations were further developed. Furthermore, we thank the ECMWF for providing access to the ERA5 reanalysis data. We are also grateful for fruitful discussions with A. Osso. Finally, we thank the Austrian Science Fund (FWF) for funding the work; the wind analysis is part of the FWF stand-alone project Strato-Clim (grant number
P-40182).

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
