# Peer review of "Closing the gap in the tropics: the added value of radio-occultation data for wind field monitoring across the equator"

_Atmospheric Measurement Techniques, 2023_

## Author Comment (AC1)

**Author's Response to Referee #1 Comments**

We would like to thank the reviewer for the constructive assessment, for generally finding our study interesting and well designed, and for the helpful comments for further improvement. We carefully considered and answered all comments below (comments quoted in *italic with gray background*, with the answers below each comment). Furthermore, for clarity and as found appropriate, we included initial versions of revised formulations, to indicate in which way we plan to improve the respective explanations in the revised manuscript.

**Referee #1 (Specific comments and questions)**

*1) Equations 1 and 2: what, precisely, are the variables x and y? I would like to see a level of detail here corresponding to that provided for Equations 3 and 4. Also, latitude is used in Equations 1 and 2 before it is introduced in association with Equations 3 and 4.*

Thank you for noticing. We will correct this and make the formulations of these Equations consistent.

*2) RO data retrievals: The key variable used in the study is geopotential height as a function of pressure (or pressure as a function of geopotential height). How is pressure retrieved? It is mentioned that the RO geopotential climatologies are available from 1000 hPa to 5 hPa. That covers atmospheric regions where the "dry" approximation is applicable as well as regions where it is certainly not applicable. Some explanations of how that is handled is needed.*

Thank you for asking. Briefly summarized, based on the atmospheric bending of the GNSS signals during the occultation sounding, it is possible to retrieve atmospheric refractivity profiles via an Abel transform. From these, air density and pressure profiles as a function of altitude, or geopotential height, can be accurately derived based on the refractivity equation, the equation of state, and the downward integration of the hydrostatic equation. Along the retrieval chain we derive the physical atmospheric parameters (e.g., physical pressure) using a moist-air retrieval described in detail in Li et al. (2019) . Roughly down to 700 hPa, but in practice significantly depending on the amount of water vapor, the physical information is dominantly dependent on information from RO measurements. This - of course - varies somewhat depending on the geographical latitude (dry polar troposphere versus moist tropical troposphere). Towards lower tropospheric altitudes, the moist information more strongly relies on background information.

We will add as additional reference for the moist-air retrieval:
Li, Y., Kirchengast, G., Scherllin-Pirscher, B., Schwaerz, M., Nielsen, J. K., Ho, S. P., & Yuan, Y. B. (2019). A new algorithm for the retrieval of atmospheric profiles from GNSS radio occultation data in moist air and comparison to 1DVar retrievals. *Remote Sensing*, *11*, 2729, https://doi.org/10.3390/rs11232729

Furthermore, we will add the following lines to Section 3.2, line 153 (initial formulation, may be refined):
"The WEGC OPSv5.6 retrieval system processes the atmospheric parameters as a function of altitude or geopotential height, based on the refractivity equation, the equation of state, and the downward integration of the hydrostatic equation. The physical atmospheric parameters (e.g., physical pressure)

are derived using a moist-air retrieval algorithm, which combines the individual profiles with background information by optimal estimation; see Li et al. (2019) for details."

**3)** *You mention that the monthly-mean RO data at the 2.5x2.5 degree grid points are computed by "Gaussian latitude-longitude weighting" within a radius of 600 km. What is the width of the Gaussian? Is it 600 km? Or is 600 km the distance from the grid point within which the profiles contributes to the grid point mean?*

Thank you – we will carefully check the description of the methodology and revise it accordingly. In general, the 600 km corresponds to the distance from the grid point within which the profiles contribute to the grid point mean. The profiles are weighted according to their longitudinal and latitudinal distance to the center with a gaussian function, with a standard deviation of 300 km (lon) and 150 km (lat).

We also intend to add in the description the following citation (in proper formatting, of course): Ladstädter et al., OPAC-IROWG 2022 conference, Talk on gridding strategies, Seggau, Austria, September 8, 2022.

**4)** *You mention the need to further average to a 5x5 degree grid for the equatorial-balance calculation. Did you try other differencing techniques than forward finite-differences? It may be to simplistic, and other differencing schemes may be more suitable.*

Thank you for this thought. In the very first step of our analysis, we tested different differencing techniques (forward, central, backward) and found only minor impact on the final wind-components. However, we will take up your suggestion and perform further tests on the differencing techniques, and accordingly improve the relevant discussion in the methodology section.

**5)** *In Section 4, the analyses and discussions related to the RO data are focused on three atmospheric layers: 10 hPa, 50 hPa, and 200 hPa. However in Figures 5 and 6, RO data down to 1000 hPa is shown. Whether it makes sense to show RO data in the lower troposphere depends on how the RO data were retrieved. Depending on the answers to comment 2 above, you should consider not to show the full vertical span down to 1000 hPa.*

Since we are using a moist-air retrieval, we consider it, in principle, not a problem to show the results in Figure 5 down to 1000 hPa. Of course, in the troposphere the data become significantly influenced by background information (as briefly described in the answer to comment 2 above). To illustrate the vertical structure of the wind components, we still prefer to keep the range in these figures down to at least 800 hPa, i.e., throughout the free troposphere down to the boundary layer. However, we will make sure to better emphasize that the focus in our analysis is not on the lower to middle troposphere, but rather on the upper troposphere and lower stratosphere.

6) *Related to comment 5, there is a sentence in Section 4.3 which I don't know how to interpret (lines 259-260): "the larger influence of moisture leads to a higher need of background information in the RO retrieval chain, and as a consequence to an increase in the bias". Is this an indication that you use the "dry" solution all the way down to 1000 hPa?*

We are using a moist-air retrieval, see also answers to questions 2 and 5. We intend to rephrase the sentence in question along these lines (initial formulation, may be refined):

"… This feature clearly relates to the core region of high-quality RO data, which is in the upper troposphere and lower stratosphere. With decreasing altitude and therefore increasing moisture content, the retrieval of atmospheric parameters relies increasingly on background information (e.g., Li et al., 2019). The RO information dominates between about 8 km to 35 km in the tropics (e.g., Scherllin-Pirscher et al., 2011). "

---

## Author Comment (AC2)

**Author's Response to Referee #2 Comments**

We thank the reviewer for the careful assessment and, despite the number of comments and reservations raised, for thinking that the outcomes of the study could be useful for the scientific community, as well as for the many helpful comments for further improvement. We carefully considered and answered the comments below, starting out with a statement to the overall "Conclusions" of the reviewer (comments quoted in *italic with gray background*, with the answers below each comment).

As part of some answers, as found appropriate, we also included initial versions of revised formulations, to indicate in which way we plan to improve the respective explanations in the revised manuscript. Furthermore, several answers are in an initial form and will be further consolidated along with the actual manuscript revision. Before we start the latter, we plan to seek initial feedback from the editor as to whether our intended revisions are deemed appropriate; in line with the AMT editorial procedures (key quotes: "…submission of a revised manuscript…is encouraged only if you can satisfactorily address all comments… In case of doubt, please ask the handling associate editor directly whether they would encourage submission of a revised manuscript…").

**Referee #2 – Conclusions:**

*Based on the limited innovations of this paper, some potential flaws in the methodology, and very limited discussion, I suggest a major revision of the paper. In the revised version, the authors should properly address the below mentioned issues. The extra analysis can be added in the supplement or appendix in order not to lengthen the paper. In the current form, the study raises more questions than it undoubtedly answers. Without performing a much more detailed analysis, I find the paper unsuitable for publication in EGU AMT. I am mostly concerned about the innovations of this paper – it seems we have not learned much, or that the authors have only confirmed what is known already. But I still do think the outcomes of the study could be useful for the scientific community.*

**Introductory author's statement (as a general answer, aiming to clarify the intended scope of the study):**
First of all, we would like to explicitly thank the reviewer for this very thorough assessment of our manuscript, which will clearly contribute to substantial improvements in the revised manuscript. We appreciate the invested time and effort and think that many of the suggestions and editorial comments are helpful indeed. Furthermore, we also plan to carefully re-check, and partly refine, several of the computations and evaluations, since we agree it is important to have a more in-depth basis for some of the results shown and discussed (for example, and in particular, related to what the added value of the v-component, i.e., the meridional wind estimations, actually may be).

This being said, we want to emphasize that the main goal and focus of this study is to test the utility of radio occultation data for wind field monitoring across the equator. Outside the equatorial band in focus here, the utility was analyzed in the "sister study" by Nimac et al. (2023), and we aim to close this remaining gap across the equator. We do have evidence that RO-derived wind fields have a clear potential to provide an "added-value", with focus on long-term wind field monitoring based on monthly mean fields at meso-scale resolution, where its unique combination of high accuracy and long-term stability (=multi-year to multi-decadal stability) can play out. This is where also the complementary value of re-processed RO-based climate wind field data records may sit, while fully acknowledging that for "weather variability" (sub-monthly timescales of hours and days) the value of the "wind information content" in RO data will best unfold via data assimilation into NWP and other atmospheric

(re)analysis systems. We hence think that this study, accepting its (limited but clear) scope and focus, does definitely have value to the RO-related readership of AMT, but also to the broader atmospheric scientific community reading AMT, since to our knowledge this type of analyses has not been done and published so far. The paper therefore has a clear methodological focus (and we will improve the introduction, as the reviewer suggested, to make this focus and scope quite clearer at the outset), which is why we submitted it to AMT, and in particular to an AMT Special Issue that focuses on RO-related studies. Clarifying this focus, we admit and agree, that we do not focus on analyzing or describing at the same time atmospheric dynamical processes as such; this is not within the scope of this study (while we confirm, well spotted by the reviewer, that we need to improve on how we present and describe the issues around the meridional wind component).

Specifically, the retrieval of RO climatic wind field data is still a quite new topic within the RO community and basics of the study were presented at the International Radio Occultation Working Group Workshop 2022, which is why we think the study also fits well in this AMT Special Issue; and the interest in this topic is available in the RO community. Once the utility of RO-based wind field datasets is established, atmospheric dynamics aspects, with focus on multi-year to decadal variability and changes, can and will be the focus of future studies. We emphasize this, since we plan to keep the scope of this study as currently laid out, i.e., we do not intend to much expand the discussion of aspects in section 4.3 towards broader application aspects. However, we are indeed thankful to the reviewer for pointing out several important issues of re-checks and evaluations we anyway will have to do for the revised manuscript within the methodological scope. Please see more details in the answers below on how we plan to implement these revisions.

**Major Comments:**

*1) In the introduction, the authors state "In this study [you] aim to close the gap in RO wind field computation across the equator". However, I am not able to easily identify this gap based on the introduction you provided. Therefore, I suggest that the authors clearly and directly identify this gap. What are the innovations that this study addresses, how do you aim to expand the present knowledge, why is the potential new knowledge important, can we use it in NWP or climate science, etc.? What new can we learn from this study in comparison to other studies? Please, elaborate in more details in the revised manuscript.*

Please, see introductory statement above, which refers to those questions. Furthermore, we will emphasize these issues more strongly in the introduction.

*2) Throughout the paper, the WMO thresholds for data quality of winds (+- 2 m/s and +- 5m/s) are mentioned. It would be nice to elaborate to which of the following data do the thresholds apply:*

   *Instantaneous winds*

   *Instantaneous zonal-mean winds*

   *Monthly-mean winds (applied in Nimac et al., 2023)*

   *Monthly-mean zonal-mean winds (e.g., Fig. 2)*

   *Monthly-mean zonal-mean latitudional-band-mean winds (e.g Fig. 3)*

*something else?*

*In the paper, you are mentioning the threshold for different of these options, but the thresholds are not equivalent for e.g., monthly-mean winds and monthly-mean zonal-mean winds, they are certainly more strict for the latter.*

Thank you for this reference. We know and acknowledge that WMO also provides more detailed and differentiated requirements, for different spatial and temporal resolutions, as well as for different applications. In general, we focus here on climate-related winds, with a fairly strong spatial and temporal averaging. For these monthly-mean winds with a horizontal averaging around 300 km, we use the 2 m/s requirement as the indicative threshold (we may possibly drop the 5 m/s threshold). Furthermore, we found in the study of Nimac et al. (2023) that a specifically given threshold has the tendency to be exceeded rather fast/abrupt at a specific geographical location, making not a huge difference whether we use an indicative threshold of, for example, 1 m/s or 2 m/s. We note that the advantage of RO-based long-term wind records is their unique potential of being also temporally stable, which is another WMO requirement of stability. That is, if we consider monthly winds with accuracy within ±2 m/s, this would be consistent with a decadal stability of roughly ±0.5 m/s per decade, which is the associated WMO-based requirement we use to evaluate long-term stability.
Having said this, we will improve the discussion and reasoning for which indicative thresholds we see fit for our study from the portfolio of requirements of the WMO.

**3)** *Furthermore, there are inaccurate claims at different instances in the introduction, which need to be revised, in relation to the references pointing to not yet revised studies.*

*For example, the study Nimac et al. (2023), in revision at the same journal, does reproduce ERA5 monthly and zonal-mean geostrophic winds rather well (their Fig. 6). It is very important to state it precisely, as suggested by the underlined text above.*

*On the other hand, Fig. 7 in Nimac et al. demonstrates that the monthly mean ROg-ERAg winds (without zonal averaging) often exceed +- 2 m/s bias threshold. Comparing their Figs. 6 and 7, it is also clear that +- 2 m/s threshold is often only achieved in the zonal-mean monthly-mean winds due to compensating biases along the latitude circle.*

We will carefully check the formulations again and revise if necessary.

**4)** *The computation of the geostrophic winds is very sensitive to the applied resolution of the input data, as you have shown in Figure 1. To avoid the zig-zag pattern at high-resolution, the authors should either use higher-order symmetric approximation of the derivatives (instead of first order forward) or compute the derivatives exactly using a spectral method. At least, the authors should prove that the choice of numerical approximation don't play a major role in the zig-zag pattern. Furthermore, I would be curious to see, how the choice of averaging period affects the "optimal" resolution (only briefly mentioned in line 165). I guess 0.5-degree resolution would not be an issue, if the data averaging was 3 months instead of a single month, but I am eager to see your results. On the other hand, I ask what the reason is for testing equatorial balance in higher-resolution reanalysis data, if the RO data are only available at 2.5-degree resolution.*

In general, we tested different formulations of the derivative operator. We tested forward, backward and central difference, and the differences in the results were essentially negligible. However, we want to take up your suggestion and calculate other approximations of the derivative, to see if it has an impact on the results. We will discuss the outcome in the method section.
About the suggestion of a higher, seasonal-temporal averaging, with a finer spatial resolution. In a short answer, we quickly tested it, and it made no fundamental difference, leading to a zig-zag pattern similar as in Figure 1.

However, we emphasize that in this study the aim is to derive monthly wind data for climate analysis and climate monitoring, following the line of work from Nimac et al. (2023). In the future the goal is to derive for the full available time-series, monthly 2.5 x 2.5 wind products (roughly from 2006 onwards).

A higher spatial resolution is on the one hand not recommendable for RO data and this time frame. This would require an even higher daily occultation statistics, which is not the case at the moment (see also Angerer et al. 2017, Ladstädter et al. 2023). On the other hand, as a further physical reason, the geostrophic and equatorial balance will also not hold well at higher temporal or spatial resolution, leading to larger ageostrophic contributions (see also the reviewer's thoughts in comment 5 below). Depending on our re-check results we consider also to drop the 0.5° resolution case in Figure 1, since it is a sub-degree-scale that is not part of the RO-based wind field records we aim at (for which no finer horizontal resolution than 300 km is thought).

*5) The equatorial balance equation for the zonal wind works reasonably well in the stratosphere in the equatorial area, but we know this already from other studies, e.g., Healy et al., 2020. The meridional wind deduced from equatorial balance equation does not seem to reproduce the original winds, as shown in Fig. 2d,e and Fig. 7b,h,e. The explanation why it fails is speculative and unconvincing ("This could be because the v component contains a derivation with respect to latitude as well as longitude which is computationally not as robust as the second derivative with respect to latitude."). Apparently, the balance is not satisfied in the deep tropics.*

*Another possible reason is that the steady-state assumption (neglecting temporal derivatives of meridional wind) might not be valid for meridional wind component. As this is one of the key results of this study, the authors should do more effort to analyse and explain it. You could do this by inspecting the magnitude of the terms in the meridional derivative of full Euler equation for meridional wind.*

*Is the inability of equatorial balance equation to reproduce meridional winds also the reason why other authors opted not to use it? I also find it rather disturbing that the analysis of meridional wind was only performed for a certain longitudinal band, - 10 to 10 degrees longitude? Why not performing similar analysis also for other bands?*

First of all, we would like for your insightful feedback, which will help to improve the manuscript. We already tested the meridional wind component on a zonal-mean average (which is close to zero), as well as for different longitude sectors. In the reviewed manuscript, we just showed the results exemplary for the longitude sector around the prime meridian (−10◦ to 10◦ longitude), keeping in mind that the other sectors were also studied and qualitatively showed similar behavior (we acknowledge that this was not made sufficiently clear).

But we agree with your comment that we need to more carefully re-assess the analysis of the meridional wind component, to see how the equatorial balance holds, and where the "added value" can be isolated (if it exists). Dependent on these re-check results we will decide on the way of how we (may) include the meridional wind component in the revised manuscript. Even if we would find it is not working well, it will be important information at least to the RO community, since several (unpublished) discussions linger around on what the "added value" possibly is.

*6) I like the results presented in Section 4.3. These are very interesting, and the revised paper should build on that, while presenting a detailed analysis why the geostrophic approximation provides an even better reconstruction.*

Thank you – we also see this as a specifically interesting result. However, a deeper physical analysis is beyond the scope of this specific study (see our introductory statement as the answer to the "Conclusions" section above).

*7) Descriptions in the figure captions should be more accurate, and English should also be improved at many places.*

Thanks, we will revise figure captions and aim to improve the writing.

**Specific comments:**

*2-3: Without "availability". Consider the following reformulation:*

*Greater availability of wind data is particularly needed, especially in tropical regions and the southern hemisphere.*

We will revise it accordingly.

*9-10: what do you mean by "volatile in derivation"? Please, express it more clearly.*

We will rephrase to:
"From analyzing first the zonal and meridional wind component, we find that the meridional wind component is more sensitive and vulnerable in the numerical implementation, however the total wind speed benefits from a computation of both wind components."

*20: Bauer et al. is not a good reference in this context, as it only briefly mentions what is missing in the observing system, but does not actually provide any content. Instead, I suggest citing Baker et al., 2014.*

Thank you for this suggestion.

*31: I would exclude AMVs here as they are almost global*

Thank you for this information.

*33: ADM Aeolus does not really perform 3D wind profiling as it only measures a profile of a projection of the wind perpendicular to the satellite track, which is quite similar to the zonal wind component.*

*32-34: This needs to be reworded. Not only that Aeolus "has potential", but it has also demonstrated its usefulness, which has been described in several studies, such as Rennie et al., 2021, Pourret et al., 2022*

Yes, thank you for this information. We will rewrite the sentence accordingly.

*30-35: I think it is important to mention that much of the wind information is nowadays obtained also implicitly in NWP to initialise the forecast, i.e. through 4D-Var humidity and/or ozone tracing (Geer et al., 2018 ; Zaplotnik et al., 2023), as well as through the geostrophic adjustment, and directly through the background-error covariances, especially where the geostrophic balance applies. The microwave humidity sounders are now the most important observation system in ECMWF IFS, in large part due to aforementioned tracing effect.*

Yes, we will do so.

*47-52: It would be informative to mention the horizontal resolution as well, not just the vertical resolution. It could also give reasoning for my further comment line 78.*

*50-51: the so-called sweet spot for GPSRO is 10-32 km, see Semane et al., 2022, their Fig. 1.*

We will add the information about the horizontal resolution in line 50. Furthermore, we will also rephrase the sentence regarding the vertical core region of RO in the following way (initial formulation, may be refined).
"RO data cover well the complete stratosphere, with a core region of high quality in the upper troposphere and lower stratosphere (Zeng et al., 2019; Steiner et al., 2020), having a typical horizontal of 200 km to 300 km (e.g., Kursinski et al. 1997, Foelsche et al. 2011)."

Foelsche, U., S. Syndergaard, J. Fritzer, and G. Kirchengast
Errors in GNSS radio occultation data: relevance of the measurement geometry and obliquity of profiles
Atmos. Meas. Tech., 4, 189-199, doi:10.5194/amt-4-189-2011, 2011

*64-65: It is important to mention that Healy et al. (2020) applied equatorial balance equation only in the stratosphere, using zonally and monthly averaged data (for apparent reasons). It is not clear, whether such balance holds also instantaneously at particular location and time instance.*

Yes, we agree that in the deep tropics the balance probably does not hold. We will carefully re-asses the meridional wind component (see major comment 5), and revise the manuscript based on the results.

*70: I would exclude "going further towards equator than other studies", as this might not be entirely justified by results in their Figs 6 and 7.*

Yes, we will do that.

*78: what is the reasoning for the choice of 2.5deg x 2.5 deg grid for the assessment of the quality of the approximation? Is it done to follow Nimac et al., 2023, or is there any physical reasoning, e.g. the horizontal resolution of the RO data? If so, it has to be explicitly written to avoid speculation. Note that by increasing the resolution, the greater portion of the total wind is represented by ageostrophic motions, which are unbalanced.*

*86: The magnitude of ageostrophic contributions are vastly influenced by the resolution at which one performs the analysis. See for example the study of Bonavita (2023), their Fig. 5.*

Thank you for your consideration. Yes, we used the same approach as in the study of Nimac et al (2023). The goal in the future is to produce a long-term monthly RO wind product on a 2.5 x 2.5 grid. Another reason is that a higher spatial and temporal resolution is not feasible for RO data. Finally, providing a physical reason, finer resolutions (temporal and spatial) will increase the ageostrophic contributions (see also answer in major comment 4).

*100-110: It appears a bit strange, that you use derivative over (x,y) in equatorial balance equation and (lambda,phi) in geostrophic balance equation. Choose one set of variables for both.*

Yes, we can change the formulation in the paper.

*125: do the WMO-OSCAR, 2023 requirements apply to instantaneous winds, monthly means or monthly and zonal-means? This is very important.*

Please refer to our answer to major comment 2. Furthermore, please note that we don't refer to instantaneous winds, and focus on monthly climate related winds.

*131: "to limit the length of the paper" is a rather strange argument. You can always provide a supplementary file in the EGU Journals.*

We will rephrase that: January was chosen as a representative month for the plots. The other months were analyzed and show no fundamental different behavior.

142: I do not agree with that statement, as mentioned in the General comments.

Please refer to our answer to major comment 4.

*Figure 1: are the zig-zag features similar at other latitudes?*

*Figure 1: it should be mentioned in the figure caption what the dashed lines represent*

Figure 1 shows the zonal component computed using the equatorial-balance equation. At latitudes north and south of 40° the zig-zag feature gets smaller. The regions outside of the equatorial region are not shown in the figure, because they are not the focus of this study, since we specifically analyze the equatorial-balance equation across the equator.
Thank you for noticing. We will add in the figure caption the description of the dashed lines, which represent ±2 m/s and ±5 m/s thresholds.

*154: does it mean that no correction due to latitudinally varying centrifugal force is applied?*

Yes, we focus on geostrophic balance.

*161-162: is 600 km the halfwidth of the Gaussian or is this the localisation threshold? If so, what is the halfwidth of the Gaussian smoother?*

*163: the smoothing procedure is rather strange – first you do a Gaussian smoother, then you further perform binning. Can you provide an example in the supplementary, how the raw fields evolve in your preprocessing routine.*

Thank you, we will carefully check the description of the methodology and revise it accordingly. In general, the 600 km corresponds to the distance from the grid point within which the profiles contribute to the grid point mean. The profiles are weighted according to their longitudinal and latitudinal distance to the center with a gaussian function, with a standard deviation of 300 km (lon) and 150 km (lat).

We also intend to add in the description the following citation (in proper formatting, of course):
Ladstädter et al., OPAC-IROWG 2022 conference, Talk on gridding strategies, Seggau, Austria, September 8, 2022.

The smoothing procedure was performed due to the fact that we used preprocessed RO data from Nimac et al. (2023), as we wanted to be as comparable as possible. When working with RO data and the equatorial balance equation, it became clear that 2.5°x2.5° is not feasible. Further binning improved the quality of the reproduced wind field.

*193: I would not say "it is not that well reproduced", I would say it is not reproduced at all (Fig 2. d,e). Given the large relative differences between v_o and v_eb, I would suggest to add a new figure of relative differences. Based on Fig 2d,e, I also find it very unconvincing to use equatorial balance for meridional wind component at all.*

*193-194: I find the explanation for the mismatch between v_eb and v_o rather unconvincing. I would say that the derived physical balance does not apply for meridional wind. If you look at the derivation precisely, there is an important assumption of steady state flow. However, the tropical disturbances are not steady, especially the features involving meridional flow such as MRG waves.*

*195-196: this might be coincidental. What is the reason for better V_eb if it contains wrong v_eb? How can it be shown?*

Please refer to the answer in major comment 5. In a first analysis the total wind speed improved, but we see a further cross-check of the meridional wind component is necessary. Based on the results, we will see if a longitudinally resolved meridional wind component will provide an added value to the total wind speed.

*207: tropopause in the deep tropics is rather found between 100 hPa and 70 hPa, instead of 200 hPa*

Thank you for this comment. In principle we follow the study of Nimac et al. (2023), using the same respective three levels. But we will rephrase and write:
"Our focus lies on the three representative levels, 200 hPa, 50 hPa and 10 hPa, relating roughly to the upper troposphere, lower stratosphere and middle stratosphere."

*213- : It is important to note that the similarity between ERA5 v_eb and RO v_eb does not imply that the use of equatorial balance is meaningful due to large differences in ERA5 v_o and v_eb. It only suggests that the input geopotential data of ERA5 and RO for the computation of u_eb and v_eb are similar. This is not unexpected, as the same COSMIC data were assimilated (albeit in a somewhat different form) in the production of ERA5 reanalysis (Hersbach et al., 2020).*

We agree with your statement in the first sentence. However, this was not our intended message in the manuscript. In analyzing the difference between *ERA5* v_eb and RO v_eb, we aim to study the systematic difference between the two data sets, as emphasized in Table 1 and the description in the main text. However, before that, we study in a first step, the bias resulting from the equatorial balance approximation, based on the state-of-the-art reanalysis ERA5 data (e.g., Figure 2); RO does not play any role for this estimation.
Apart from this, we agree that ERA5 has in general RO data assimilated, and hence, is not independent of RO. However, since all major (re)analyses do assimilate RO data since 2006 (start of the "U.S. COSMIC" and "European Metop" RO multi-satellite era), we consider it adequate in this study to quantitatively evaluate the equatorial balance approximation using RO data, and comparing it with the wind field data of the state-of-the-art reanalysis ERA5. From other previous studies that also involve short-range forecasts, or MERRA2, and JRA-55 reanalyses, like for example in the study from von Schuckmann et al. 2023 https://doi.org/10.5194/essd-15-1675-2023, Sect. 3 therein, we know that this will likely result in no major differences, and should be sufficient for the present purpose. Furthermore, the approach of the two-step analysis, as described in Table 1, exactly decomposes the analysis into the bias from the approximation (first step, only ERA5), and the bias between the two data set (ERA5 and RO), aiming to partially circumvent this specific problem.

*224: provide references to those missions.*

We already provided references in the paragraph from line 155 to 160.

*231-233: The alignment of an increase of systematic bias with the drop in the number of RO profiles is a very interesting feature. However, my question is how you can be certain that only this factor explains the increase of bias. No proof is provided, so the statement should be milder and speculative. From the statistical perspective, a reduction of the number of profiles would only increase the random error.*

Yes, we will revise the sentence with a more conservative formulation. Furthermore, we cite the work about the uncertainty propagation in RO retrieval from Schwarz et al. (2017).

*Figure 7 is another proof, that v_eb (as well as v_g) are likely unable to approximate v_o.*

Please refer to major comment 5.

*276-277: as this is not some new conclusion, I would say "as in Healy et al. (2020)".*

We will mention in that statement the important initial pre-work of Healy et al. (2020). Nevertheless, we see this work here as a more detailed analysis of the wind fields across the equator, providing further insight about the potential of RO data for wind field monitoring.

*277: what do you mean by "the resolution was possible to obtain" (I could not understand with going back to the results section). Please, express more clearly.*

Yes, we will rephrase. Please see also major comment 5.

*1: vertically*

*7-8: sentence "We analyze the equatorial balance equation within this latitude band." Is redundant in my opinion.*

*30: several heights but mostly upper troposphere*

*61: no comma before "isobaric levels".*

*63: between 15N in 10 S.*

*65: analyzed instead of "started to analyze"*

*66: to reproduce ERA5 geostrophic winds ("original winds" sound like total winds). You properly introduce "original" only later in the text, in line 115, leaving the reader confused at this stage.*

*68: I am not sure whether "Anthes" region is an established geographical term. Did you perhaps mean Andes?*

*70: equatorial band*

*70: "approaches" instead of "converges"*

*71: Reformulate sentence "Interesting was also to see…"*

*79: latitude-longitude*

*97-104: it is necessary to mention, that Coriolis parameter is now approximated using equatorial beta-plane approximation.*

*106: remove "still"*

*135: "includes"/"provides" instead of "combines"*

*135 and 137: sometimes you use "data" as singular noun and sometimes as plural, e.g. "The ERA5 reanalysis data combines…" vs. "The data are available…"*

*138: no comma before "to find", no comma before "for the equatorial balance…" Revise misuse of comma at the other places of the text as well.*

*178: no comma*

*Figure 4 should include two more rows: 1) monthly-mean winds V_o and 2) monthly-mean winds V_eb. The caption should be: Temporal development of the wind-speed bias…*

*222: revise "high occultation statistics"*

*237: "section"*

*237: no comma before "to complete"*

*244-245: "geostrophic break down" to the "the geostrophic approximation does not apply any more"*

*272: it is again unclear to the reader, what are the "winds calculated using ERA5 data and original winds". Try forming the Conclusions in a way that is understood even to readers who did not read the whole methodology*

*276: word order: "we could successfully apply"*

*287: first comma is excessive*

*279: this reads as the zonal wind speeds are 1 m/s and meridional wind speeds are 15 m/s. Again, be more precise for which levels in the equatorial +-5 deg channel do this wind speeds apply.*

Thank you for noticing! All corrections will be done.

References:

Baker, W. E., and Coauthors, 2014: Lidar-Measured Wind Profiles: The Missing Link in the Global Observing System. Bull. Amer. Meteor. Soc., 95, 543–564, https://doi.org/10.1175/BAMS-D-12-00164.1.

Bonavita, 2023: On the limitations of data-driven weather forecasting models. ArXiv:2309.08473

Geer, AJ, Lonitz, K, Weston, P, et al. All-sky satellite data assimilation at operational weather forecasting centres. Q J R Meteorol Soc. 2018; 144: 1191–1217. https://doi.org/10.1002/qj.3202

Rennie, M.P., Isaksen, L., Weiler, F., de Kloe, J., Kanitz, T. & Reitebuch, O.(2021) The impact of Aeolus wind retrievals on ECMWF global weather forecasts. Q J R Meteorol Soc, 147(740, 3555–3586. Available from: https://doi.org/10.1002/qj.4142

Pourret, V., Šavli, M., Mahfouf, J.-F., Raspaud, D., Doerenbecher, A., Bénichou, H., et al. (2022) Operational assimilation of Aeolus winds in the Météo-France global NWP model ARPEGE. Quarterly Journal of the Royal Meteorological Society, 148(747), 2652–2671. Available from: https://doi.org/10.1002/qj.4329

Semane, N., R. Anthes, J. Sjoberg, S. Healy, and B. Ruston, 2022: Comparison of Desroziers and Three-Cornered Hat Methods for Estimating COSMIC-2 Bending Angle Uncertainties. J. Atmos. Oceanic Technol., 39, 929–939, https://doi.org/10.1175/JTECH-D-21-0175.1.

Zaplotnik, Ž., Žagar, N. & Semane, N.(2023) Flow-dependent wind extraction in strong-constraint 4D-Var. Quarterly Journal of the Royal Meteorological Society, 149(755, 2107–2124. Available from: https://doi.org/10.1002/qj.4497

Citation: https://doi.org/10.5194/amt-2023-137-RC2

---

## Author Response (AR1)

**Collective Author's Response to Referee #1 and Referee #2**

We thank the reviewers for their careful and constructive assessment, for generally thinking that the outcomes of the study could be useful for the scientific community, as well as for the many helpful comments for further improvement. We carefully considered all points raised, substantially revised and improved the manuscript along these lines, and provide answers to the comments below (comments quoted in *italic with gray background*, with the answers below each comment).

**Author's Response to Referee #1 Comments**

**Referee #1 (Specific comments and questions)**

*1) Equations 1 and 2: what, precisely, are the variables x and y? I would like to see a level of detail here corresponding to that provided for Equations 3 and 4. Also, latitude is used in Equations 1 and 2 before it is introduced in association with Equations 3 and 4.*

Thank you for noticing. We corrected this and made the formulations of the equations consistent.

*2) RO data retrievals: The key variable used in the study is geopotential height as a function of pressure (or pressure as a function of geopotential height). How is pressure retrieved? It is mentioned that the RO geopotential climatologies are available from 1000 hPa to 5 hPa. That covers atmospheric regions where the "dry" approximation is applicable as well as regions where it is certainly not applicable. Some explanations of how that is handled is needed.*

Thank you for asking. Briefly summarized, based on the atmospheric bending of the GNSS signals during the occultation sounding, it is possible to retrieve atmospheric refractivity profiles via an Abel transform. From these, air density and pressure profiles as a function of altitude, or geopotential height, can be accurately derived based on the refractivity equation, the equation of state, and the downward integration of the hydrostatic equation. Along the retrieval chain we derive the physical atmospheric parameters (e.g., physical pressure) using a moist-air retrieval described in detail in Li et al. (2019) . Roughly down to 700 hPa, but in practice significantly depending on the amount of water vapor, the physical information is dominantly dependent on information from RO measurements. This - of course - varies somewhat depending on the geographical latitude (dry polar troposphere versus moist tropical troposphere). Towards lower tropospheric altitudes, the moist information more strongly relies on background information.

We added this reference as an additional one for the moist-air retrieval:
Li, Y., Kirchengast, G., Scherllin-Pirscher, B., Schwaerz, M., Nielsen, J. K., Ho, S. P., & Yuan, Y. B. (2019). A new algorithm for the retrieval of atmospheric profiles from GNSS radio occultation data in moist air and comparison to 1DVar retrievals. *Remote Sensing*, *11*, 2729, https://doi.org/10.3390/rs11232729

Furthermore, we added the following text insert in Section 3.2, line 153:
"The WEGC OPSv5.6 retrieval system processes the atmospheric parameters as a function of altitude or geopotential height, based on the refractivity equation, the equation of state, and the downward integration of the hydrostatic equation. The physical atmospheric parameters (e.g., physical pressure) are derived using a moistair retrieval algorithm, which combines the individual profiles with background information by optimal estimation; see Li et al. (2019) for details."

**3)** *You mention that the monthly-mean RO data at the 2.5x2.5 degree grid points are computed by "Gaussian latitude-longitude weighting" within a radius of 600 km. What is the width of the Gaussian? Is it 600 km? Or is 600 km the distance from the grid point within which the profiles contribute to the grid point mean?*

Thank you, we improved the description of the methodology. In general, the 600 km corresponds to the distance from the grid point, defined as the center location of the area of influence, within which the profiles contribute to the grid point mean. In performing the averaging, the profiles are weighted according to their distance from this center location with a bivariate (latitude-longitude) gaussian function which peaks at the center and features a standard deviation of 150 km along latitude and 300 km along longitude, respectively.

We also added the following citation:
Ladstädter et al., OPAC-IROWG 2022 conference, Talk on gridding strategies, Seggau, Austria, September 8, 2022.

**4)** *You mention the need to further average to a 5x5 degree grid for the equatorial-balance calculation. Did you try other differencing techniques than forward finite-differences? It may be too simplistic, and other differencing schemes may be more suitable.*

Thank you for this thought. In the very first step of our analysis, we tested different finite-differencing techniques (centered, forward, backward, and centralized with higher-order). We found that while forward and backward differencing is not recommendable, centralized and higher-order centralized methods show very similar results when using ERA5 data on a 2.5° x 2.5° grid. The local approximation bias at individual grid points ($V_{EB} - V_O$) is slightly smaller when using the standard central method, while the zonal mean bias improves a bit with the higher-order method. These biases are amplified when using the RO data available on a 5° x 5° grid. Here the difference in the local bias is found larger, with the standard central method outperforming the higher-order method. This larger local bias of the higher-order 5-point method compared to the standard 3-point method is likely caused by the fairly large latitudinal range of the former across the central grid point, spanning across four 5° steps. For the zonal-mean bias, again the higher-order method performs somewhat better, with the quality depending on altitude level and month. Overall, since the equatorial balance approximation is, strictly speaking, only fully valid at the equator, the approximation error from including data points outside of the ±5° equator band is considered larger than the gain from applying the higher-order method. For this reason, the standard centered differencing method was finally chosen as the primary method for the respective data analyses in this study.

**5)** *In Section 4, the analyses and discussions related to the RO data are focused on three atmospheric layers: 10 hPa, 50 hPa, and 200 hPa. However in Figures 5 and 6, RO data down to 1000 hPa is shown. Whether it makes sense to show RO data in the lower troposphere depends on how the RO data were retrieved. Depending on the answers to comment 2 above, you should consider not to show the full vertical span down to 1000 hPa.*

Since we are using a moist-air retrieval, we consider it not as a problem to show the results in Figure 5 down to 1000 hPa. Of course, in the troposphere the data become significantly influenced by background information (as briefly described in the answer to comment 2 above). Nevertheless, since our focus is the free troposphere (and

of course the stratosphere), where frictional forcing can be neglected, we will skip the atmospheric boundary layer and show the results only down to 800 hPa. We added a sentence about this in the discussion of the results. Furthermore, we will also better emphasize that the core range of RO data is from the upper troposphere to the lower stratosphere (see added sentences in the Introduction and Section 4.3).

*6) Related to comment 5, there is a sentence in Section 4.3 which I don't know how to interpret (lines 259-260): "the larger influence of moisture leads to a higher need of background information in the RO retrieval chain, and as a consequence to an increase in the bias". Is this an indication that you use the "dry" solution all the way down to 1000 hPa?*

We are using a moist-air retrieval, see also answers to questions 2 and 5. Furthermore, we rephrased the sentence in question:

"… This feature clearly relates to the core region of high-quality RO data, which is in the upper troposphere and lower stratosphere. With decreasing altitude and therefore increasing moisture content, the retrieval of atmospheric parameters relies increasingly on background information (e.g., Li et al., 2019). The RO information dominates between about 8 km to 35 km in the tropics (e.g., Scherllin-Pirscher et al., 2011)."

**Author's Response to Referee #2 Comments**

**Referee #2 – Conclusions:**

*Based on the limited innovations of this paper, some potential flaws in the methodology, and very limited discussion, I suggest a major revision of the paper. In the revised version, the authors should properly address the below mentioned issues. The extra analysis can be added in the supplement or appendix in order not to lengthen the paper. In the current form, the study raises more questions than it undoubtedly answers. Without performing a much more detailed analysis, I find the paper unsuitable for publication in EGU AMT. I am mostly concerned about the innovations of this paper – it seems we have not learned much, or that the authors have only confirmed what is known already. But I still do think the outcomes of the study could be useful for the scientific community.*

**Introductory author's statement (as a general answer, aiming to clarify the intended scope of the study):**
First of all, we would like to explicitly thank the reviewer for this very thorough assessment of our manuscript, which clearly contributed to substantial improvements in the revised manuscript. We appreciate the invested time and effort and found many of the suggestions and editorial comments helpful indeed. We also carefully rechecked, and partly refined, several of the computations and evaluations, since we agree it is important to have a more in-depth basis for some of the results shown and discussed (for example, and in particular, related to where the added value of the v-component, i.e., the meridional wind estimations, does show up specifically).
This being said, we emphasize that the main goal and focus of this study is to test the utility of RO data for wind field monitoring across the equator. Outside the equatorial band in focus here, the utility was analyzed in the "sister study" by Nimac et al. (2023), and we here aim to close this remaining gap across the equator. We do have evidence that RO-derived wind fields have a clear potential to provide an "added-value", with focus on long-term wind field monitoring based on monthly mean fields at meso-scale resolution, where its unique combination of high accuracy and long-term stability (=multi-year to multi-decadal stability) can play out. This is where also the

complementary value of re-processed RO-based climate wind field data records will sit, while fully acknowledging that for "weather variability" (sub-monthly timescales of hours and days) the value of the "wind information content" in RO data will best unfold via data assimilation into NWP and other atmospheric (re)analysis systems. We hence think that this study, accepting its (limited but clear) scope and focus, does definitely have value to the RO-related readership of AMT, but also to the broader atmospheric scientific community reading AMT, since to our knowledge this type of analyses has not been done and published so far.

The paper therefore has a clear methodological focus (and we improved the introduction, as the reviewer suggested, to make this focus and scope clearer at the outset), which is why we submitted it to AMT, and in particular to an AMT Special Issue that focuses on RO-related studies. Clarifying this focus, we admit and agree, that we do not focus on analyzing or describing at the same time atmospheric dynamical processes as such; this is not within the scope of this study (while we confirm, well spotted by the reviewer, that we needed to re-check and improve on how we present and describe the issues around the meridional wind component).

Specifically, the retrieval of RO climatic wind field data is still a quite new topic within the RO community and basics of the study were presented at the International Radio Occultation Working Group Workshop 2022, which is why we think the study also fits well in this AMT Special Issue; and the interest in this topic is available in the RO community. Once the utility of RO-based wind field datasets is established, atmospheric dynamics aspects, with focus on multi-year to decadal variability and changes, can and will be the focus of future studies. We emphasize this, since we decided to keep the scope of this study as originally laid out, i.e., we did not intend to much expand the discussion of aspects in section 4.3 towards broader application aspects. However, we are indeed thankful to the reviewer for having pointed out several important issues for re-checks and evaluations that we have done now for the revised manuscript, within the methodological scope. Please see more details in the answers below on how we implemented the revisions.

**Major Comments:**

*1) In the introduction, the authors state "In this study [you] aim to close the gap in RO wind field computation across the equator". However, I am not able to easily identify this gap based on the introduction you provided. Therefore, I suggest that the authors clearly and directly identify this gap. What are the innovations that this study addresses, how do you aim to expand the present knowledge, why is the potential new knowledge important, can we use it in NWP or climate science, etc.? What new can we learn from this study in comparison to other studies? Please, elaborate in more details in the revised manuscript.*

Please see the introductory statement above, which refers to these questions. Furthermore, we emphasized these issues more strongly in the introduction now, but also in the final summary and conclusions section.

*2) Throughout the paper, the WMO thresholds for data quality of winds (+- 2 m/s and +- 5m/s) are mentioned. It would be nice to elaborate to which of the following data do the thresholds apply:*

*Instantaneous winds*

*Instantaneous zonal-mean winds*

*Monthly-mean winds (applied in Nimac et al., 2023)*

*Monthly-mean zonal-mean winds (e.g., Fig. 2)*

*Monthly-mean zonal-mean latitudinal-band-mean winds (e.g Fig. 3)*

*something else?*

*In the paper, you are mentioning the threshold for different of these options, but the thresholds are not equivalent for e.g., monthly-mean winds and monthly-mean zonal-mean winds, they are certainly more strict for the latter.*

Thank you for this care. We know and acknowledge that WMO also provides more detailed and differentiated requirements, for different spatial and temporal resolutions, as well as for different applications. In general, we focus here on climate-related winds, with a fairly strong spatial and temporal averaging. For these monthly-mean winds with a horizontal averaging around 300 km, we use the 2 m/s requirement as the main indicative threshold. Furthermore, we found in the study of Nimac et al. (2023) that a specifically given threshold has the tendency to be exceeded rather fast/abrupt at a specific geographical location, making hence not a huge difference whether we use an indicative threshold of 2 m/s (as we chose) or a bit more tight, like 1 m/s.

The further reason for our choice is related to the aspect that the advantage of RO-based long-term wind records is their unique potential of being also temporally stable, which is another WMO requirement on stability. That is, if we consider monthly winds with accuracy within ±2 m/s, this is roughly consistent with a decadal stability of ±0.5 m/s per decade, which is the associated WMO-based requirement that we use to evaluate long-term stability (see Nimac et al., 2023).

Having said this, we improved the discussion and reasoning for the indicative thresholds we chose for our study from the portfolio of requirements of the WMO in the Method Section 2.

**3)** *Furthermore, there are inaccurate claims at different instances in the introduction, which need to be revised, in relation to the references pointing to not yet revised studies.*

*For example, the study Nimac et al. (2023), in revision at the same journal, does reproduce ERA5 monthly and zonal-mean geostrophic winds rather well (their Fig. 6). It is very important to state it precisely, as suggested by the underlined text above.*

*On the other hand, Fig. 7 in Nimac et al. demonstrates that the monthly mean ROg-ERAg winds (without zonal averaging) often exceed +- 2 m/s bias threshold. Comparing their Figs. 6 and 7, it is also clear that +- 2 m/s threshold is often only achieved in the zonal-mean monthly-mean winds due to compensating biases along the latitude circle.*

Thank you for your comment. We reformulated the sentence and hope it is clearer now, as follows:

"It was possible to reproduce the original ERA5 winds rather well, and within the target of ±2 m s$^{-1}$. However, in the region of the jet stream the difference between the two data sets exceeded this threshold. Furthermore, over large mountain areas (e.g., Himalayan or Andes region) larger deviations were found, since the ageostrophic contribution grows in importance in such regions with massive influence of topography."

**4)** *The computation of the geostrophic winds is very sensitive to the applied resolution of the input data, as you have shown in Figure 1. To avoid the zig-zag pattern at high-resolution, the authors should either use higher-order symmetric approximation of the derivatives (instead of first order forward) or compute the derivatives exactly using a spectral method. At least, the authors should prove that the choice of numerical approximation don't play a major role in the zig-zag pattern. Furthermore, I would be curious to see, how the choice of averaging period*

*affects the "optimal" resolution (only briefly mentioned in line 165). I guess 0.5-degree resolution would not be an issue, if the data averaging was 3 months instead of a single month, but I am eager to see your results. On the other hand, I ask what the reason is for testing equatorial balance in higher-resolution reanalysis data, if the RO data are only available at 2.5-degree resolution.*

Thank you for this thought. In the very first step of our analysis, we tested different finite-differencing techniques (centered, forward, backward, and centralized with higher-order). We found that while forward and backward differencing is not recommendable, centralized and higher-order centralized methods show very similar results when using ERA5 data on a 2.5° x 2.5° grid. The local approximation bias at individual grid points ($V_{EB} - V_O$) is slightly smaller when using the standard central method, while the zonal mean bias improves a bit with the higher-order method. These biases are amplified when using the RO data available on a 5° x 5° grid. Here the difference in the local bias is found larger, with the standard central method outperforming the higher-order method. This larger local bias of the higher-order 5-point method compared to the standard 3-point method is likely caused by the fairly large latitudinal range of the former across the central grid point, spanning across four 5° steps. For the zonal-mean bias, again the higher-order method performs somewhat better, with the quality somewhat depending on altitude level and month. Overall, since the equatorial balance approximation is, strictly speaking, only fully valid at the equator, the approximation error from including data points outside of the ±5° equator band is considered larger than the gain from applying the higher-order method. For this reason, the standard centered differencing method was finally chosen as the primary method for the respective data analyses in this study.

For further information see the newly introduced Appendix A.

About the suggestion of a stronger temporal averaging, seasonal rather than monthly, with a finer spatial resolution: we tested it and found no fundamental difference, leading to a zig-zag pattern similar as in Figure 1. However, we emphasize that in this study the aim is to derive monthly wind data for climate analysis and climate monitoring; see the introductory statement above that summarizes the aims of this work. In the future, the goal is to derive long-term time series, in form of monthly 2.5° x 2.5° wind products (with RO from 2006 onwards).
A finer spatial resolution is, on the one hand, not recommendable for RO data and this time frame. This would require more dense global coverage with daily RO events, which is not the available up to now (see also Angerer et al., 2017; Ladstädter et al., 2023). On the other hand, as a further physical reason, the geostrophic and equatorial balance will also not hold well at higher temporal or spatial resolution, leading to larger ageostrophic contributions (see also comment 5 below). For this reason, we now dropped the 0.5° resolution variant from Figure 1, since it is a "sub-scale" that is not relevant for the RO-based wind field records we aim at where we will (and can) not got finer than about 300 km. We have worked these arguments into the manuscript text now, at different locations.

*5) The equatorial balance equation for the zonal wind works reasonably well in the stratosphere in the equatorial area, but we know this already from other studies, e.g., Healy et al., 2020. The meridional wind deduced from equatorial balance equation does not seem to reproduce the original winds, as shown in Fig. 2d,e and Fig. 7b,h,e. The explanation why it fails is speculative and unconvincing ("This could be because the v component contains a derivation with respect to latitude as well as longitude which is computationally not as robust as the second derivative with respect to latitude."). Apparently, the balance is not satisfied in the deep tropics.*

*Another possible reason is that the steady-state assumption (neglecting temporal derivatives of meridional wind) might not be valid for meridional wind component. As this is one of the key results of this study, the authors should do more effort to analyse and explain it. You could do this by inspecting the magnitude of the terms in the meridional derivative of full Euler equation for meridional wind.*

*Is the inability of equatorial balance equation to reproduce meridional winds also the reason why other authors opted not to use it? I also find it rather disturbing that the analysis of meridional wind was only performed for a certain longitudinal band, - 10 to 10 degrees longitude? Why not performing similar analysis also for other bands?*

First of all, we would like to thank you for your insightful feedback, which significantly helped to improve the manuscript in this respect. Actually, we analyzed the meridional wind component both for zonal-mean averages (which are close to zero) and in longitude-resolved form for different longitude sectors. In the original manuscript, we just had shown the example results for the one longitude sector around the prime (Greenwich) meridian ($-10°$ to $10°$ longitude), keeping notice that we had studied the other sectors as well, which qualitatively showed similar behavior. However, we agree with your quest for a closer recheck and evaluation and now show results for all longitude sectors (new Figure 2). Furthermore, we explicitly make clear that later on we will just illustrate the prime meridian sector, as an exemplary longitude band.

Furthermore, agreeing that we needed to more carefully re-assess the detailed behavior of the meridional wind component, to see how the equatorial balance holds and where really the "added value" of this component can be isolated. For this reason, we added further new/updated figures in the revised manuscript (see Figures 2, 3, 4, 6). In addition, and in line with the updated results shown, we revised the Section 4.1 and also the Conclusions. Briefly summarizing here, the updated results include:

The meridional wind component is very small in magnitude in the tropical stratosphere and generally much smaller than the zonal wind (close to zero compared to the zonal wind speed), and its estimated equatorial-balance approximation bias is larger than its (very small) magnitude itself. For these reasons, there is no added-value information that we could derive from including the meridional component estimation in the stratosphere on top of the zonal wind estimation. However, we nevertheless clearly could show that there is demonstrable added value of including the meridional component in the troposphere, where the total wind speed estimation improves due to its inclusion on top of the zonal wind; we show and discuss this now along with Figure 3 (showing absolute and relative differences) as well as Figure 6.

**6)** *I like the results presented in Section 4.3. These are very interesting, and the revised paper should build on that, while presenting a detailed analysis why the geostrophic approximation provides an even better reconstruction.*

Thank you – we also see this as a specifically interesting result. However, a deeper physical analysis is beyond the scope of this specific study; see our detailed introductory statement as the answer to Comment 1 ("Conclusions") above.

**7)** *Descriptions in the figure captions should be more accurate, and English should also be improved at many places.*

Thanks, we revised figure caption texts and aimed to improve the English writing.

**Specific comments:**

*2-3: Without "availability". Consider the following reformulation:*

*Greater availability of wind data is particularly needed, especially in tropical regions and the southern hemisphere.*

We revised it accordingly.

*9-10: what do you mean by "volatile in derivation"? Please, express it more clearly.*

Thank you, we revised the abstract during the review process.

*20: Bauer et al. is not a good reference in this context, as it only briefly mentions what is missing in the observing system, but does not actually provide any content. Instead, I suggest citing Baker et al., 2014.*

Thank you for this suggestion, we exchanged the citation.

*31: I would exclude AMVs here as they are almost global*

Thank you for this information. We toned this down somewhat, qualified it better, and revised the sentence in the following way:
"Aircrafts and atmospheric motion vectors (AMVs) from geostationary or polar satellites provide a high temporal and horizontal sampling at several heights, but have distinct limits in accurate vertical geolocation and resolution and global representation."

*33: ADM Aeolus does not really perform 3D wind profiling as it only measures a profile of a projection of the wind perpendicular to the satellite track, which is quite similar to the zonal wind component.*

*32-34: This needs to be reworded. Not only that Aeolus "has potential", but it has also demonstrated its usefulness, which has been described in several studies, such as Rennie et al., 2021, Pourret et al., 2022*

Yes, thank you for this information. We rewrote the sentence accordingly.

*30-35: I think it is important to mention that much of the wind information is nowadays obtained also implicitly in NWP to initialise the forecast, i.e. through 4D-Var humidity and/or ozone tracing (Geer et al., 2018; Zaplotnik et al., 2023), as well as through the geostrophic adjustment, and directly through the background-error covariances, especially where the geostrophic balance applies. The microwave humidity sounders are now the most important observation system in ECMWF IFS, in large part due to aforementioned tracing effect.*

Thank you for these thoughts. We imported this aspect in form of an added text as follows:
"Wind information is nowadays obtained also implicitly as part of variational data assimilation ("4D-Var") in numerical weather prediction analyses that initialize the forecasts, such as through the geostrophic adjustment and directly through the background error covariances (especially where the geostrophic balance applies) as well as through 4D-Var of humidity and/or ozone tracing data (Geer et al., 2018; Zaplotnik et al., 2023)."

*47-52: It would be informative to mention the horizontal resolution as well, not just the vertical resolution. It could also give reasoning for my further comment line 78.*

*50-51: the so-called sweet spot for GPSRO is 10-32 km, see Semane et al., 2022, their Fig. 1.*

We added the information about the horizontal resolution in line 50. Furthermore, we rephrased the sentence regarding the vertical core region of RO in the following way.

"RO data cover well the (free) troposphere and the stratosphere, with a core region of high quality in the upper troposphere and lower stratosphere (e.g., Zeng et al., 2019; Steiner et al., 2020), having a horizontal resolution of about 200 km to 300 km (e.g., Kursinski et al., 1997; Foelsche et al., 2011)."

With the new citation Foelsche et al. (2011):

Foelsche, U., S. Syndergaard, J. Fritzer, and G. Kirchengast (2011), Errors in GNSS radio occultation data: relevance of the measurement geometry and obliquity of profiles, Atmos. Meas. Tech., 4, 189-199, https://doi.org/10.5194/amt-4-189-2011

*64-65: It is important to mention that Healy et al. (2020) applied equatorial balance equation only in the stratosphere, using zonally and monthly averaged data (for apparent reasons). It is not clear, whether such balance holds also instantaneously at particular location and time instance.*

We adapted the sentence to: "Healy et al. (2020), on the other hand, tested the zonal equatorial balance equation around the equator, studying the utility of RO data in a 5°-zonal band in the stratosphere."
For further discussion on the equatorial balance, and the added value of the meridional component, please see the answer to major comment 5.

*70: I would exclude "going further towards equator than other studies", as this might not be entirely justified by results in their Figs 6 and 7.*

Ok, we agree and deleted this part of the sentence.

*78: what is the reasoning for the choice of 2.5deg x 2.5 deg grid for the assessment of the quality of the approximation? Is it done to follow Nimac et al., 2023, or is there any physical reasoning, e.g. the horizontal resolution of the RO data? If so, it has to be explicitly written to avoid speculation. Note that by increasing the resolution, the greater portion of the total wind is represented by ageostrophic motions, which are unbalanced.*

*86: The magnitude of ageostrophic contributions are vastly influenced by the resolution at which one performs the analysis. See for example the study of Bonavita (2023), their Fig. 5.*

Thank you for your consideration. Yes, we used the same approach as in the study of Nimac et al (2023). The goal in the future is to produce a long-term monthly RO wind product on a 2.5° x 2.5° grid. The one reason is that a higher spatial and temporal resolution is not feasible given the limits of spatiotemporal sampling by RO events from the available RO missions. The other reason is more physical, i.e., finer resolutions (temporal and spatial) would increase the ageostrophic contributions (see also the answer to major comment 4). We include now an improved discussion about this in Sec. 3.1 and Sec. 3.2.

*100-110: It appears a bit strange, that you use derivative over (x,y) in equatorial balance equation and (lambda,phi) in geostrophic balance equation. Choose one set of variables for both.*

Thank you. We changed the formulation in the paper to a consistent formulation.

*125: do the WMO-OSCAR, 2023 requirements apply to instantaneous winds, monthly means or monthly and zonal-means? This is very important.*

Please refer to our answer to major comment 2. We do provide a discussion about it in Sec. 2 (which we also improved). We focus on monthly-mean climate related winds, at spatial resolution not finer than about 300 km.

*131: "to limit the length of the paper" is a rather strange argument. You can always provide a supplementary file in the EGU Journals.*

In order to clarify this point, we always analyzed all months and also show some results for the complete year 2009. Where found sufficient, we only show the results for January 2009, as a representative month. For better understanding we rephrased the relevant introductory paragraph of Sec. 3 to:
"… January 2009 was chosen as a representative month in the results section. All other months were analyzed as well and generally showed no major differences in behavior, which justifies the representative-month approach for most result discussions. As we also performed the analysis for the complete year 2009, for both ERA5 and RO data, we draw from these results to discuss aspects of seasonal and interhemispheric changes."

*142: I do not agree with that statement, as mentioned in the General comments.*

Please refer to our answer to major comment 4.

*Figure 1: are the zig-zag features similar at other latitudes?*

*Figure 1: it should be mentioned in the figure caption what the dashed lines represent*

Figure 1 shows the zonal component computed using the equatorial-balance equation. We checked at latitudes outside the equatorial band: at latitudes north and south of 40° the zig-zag features become smaller. Note that the regions outside of the equatorial region are not shown in the figure, because they are not the focus of this study; we specifically analyze the equatorial-balance equation across the equator.
Thank you for noticing. We added in the figure caption the description of the dashed line, which represents the ±2 m/s threshold.

*154: does it mean that no correction due to latitudinally varying centrifugal force is applied?*

Yes, we focus on geostrophic balance.

*161-162: is 600 km the halfwidth of the Gaussian or is this the localization threshold? If so, what is the halfwidth of the Gaussian smoother?*

*163: the smoothing procedure is rather strange – first you do a Gaussian smoother, then you further perform binning. Can you provide an example in the supplementary, how the raw fields evolve in your preprocessing routine.*

Thank you, we improved the description of the methodology. In general, the 600 km corresponds to the distance from the grid point, defined as the center location of the area of influence, within which the profiles contribute to the grid point mean. In performing the averaging, the profiles are weighted according to their distance from this center location with a bivariate (latitude-longitude) gaussian function which peaks at the center and features a standard deviation of 150 km along latitude and 300 km along longitude, respectively.

We also added the following citation:

Ladstädter et al., OPAC-IROWG 2022 conference, Talk on gridding strategies, Seggau, Austria, September 8, 2022.

About the smoothing procedure: thank you for also pointing that out. The procedure was described in an unclear manner, we apologize for that. We reformulated in the following way:

"Tests revealed that a Gaussian smoothing with a 5° longitudinal smoothing window improved the results. This smoothing was therefore applied to the equatorial-balance wind fields derived from RO data."

*193: I would not say "it is not that well reproduced", I would say it is not reproduced at all (Fig 2. d,e). Given the large relative differences between v_o and v_eb, I would suggest to add a new figure of relative differences. Based on Fig 2d,e, I also find it very unconvincing to use equatorial balance for meridional wind component at all.*

*193-194: I find the explanation for the mismatch between v_eb and v_o rather unconvincing. I would say that the derived physical balance does not apply for meridional wind. If you look at the derivation precisely, there is an important assumption of steady state flow. However, the tropical disturbances are not steady, especially the features involving meridional flow such as MRG waves.*

*195-196: this might be coincidental. What is the reason for better V_eb if it contains wrong v_eb? How can it be shown?*

Please refer to the answer to major comment 5. We completely revised this section and added the new Figures 2, 3, and 4 to show the results of our updated analyses.

*207: tropopause in the deep tropics is rather found between 100 hPa and 70 hPa, instead of 200 hPa*

Thank you for this comment. In principle we follow the study of Nimac et al. (2023), using the same respective three levels. We rephrased the sentence:

"Our focus lies on the three representative levels, 200 hPa, 50 hPa and 10 hPa, representing the tropical upper troposphere, lower stratosphere and middle stratosphere, respectively."

*213- : It is important to note that the similarity between ERA5 v_eb and RO v_eb does not imply that the use of equatorial balance is meaningful due to large differences in ERA5 v_o and v_eb. It only suggests that the input geopotential data of ERA5 and RO for the computation of u_eb and v_eb are similar. This is not unexpected, as the same COSMIC data were assimilated (albeit in a somewhat different form) in the production of ERA5 reanalysis (Hersbach et al., 2020).*

We agree with your statement in the first sentence. However, this was not our intended message in the manuscript. In analyzing the difference between ERA5 v_eb and RO v_eb, we aim to study the systematic difference between the two data sets, as emphasized in Table 1 and the description in the main text. However, before that, we study in a first step, the bias resulting from the equatorial balance approximation, based on the state-of-the-art reanalysis ERA5 data (e.g., Figure 2); RO does not play any role for this estimation.
We now added a further explanation about this in Section 4.2.
Apart from this, we agree that ERA5 has in general RO data assimilated, and hence, is not independent of RO. However, since all major (re)analyses do assimilate RO data since 2006 (start of the "U.S. COSMIC" and "European Metop" RO multi-satellite era), we consider it adequate in this study to quantitatively evaluate the equatorial balance approximation using RO data, and comparing it with the wind field data of the state-of-the-art reanalysis ERA5. From other previous studies that also involve short-range forecasts, or MERRA2, and JRA-55 reanalyses,

like for example in the study from von Schuckmann et al. 2023 https://doi.org/10.5194/essd-15-1675-2023, Sect. 3 therein, we know that this will likely result in no major differences, and is hence considered sufficient for the present purpose. Furthermore, the approach of the two-step analysis, as described in Table 1, exactly decomposes the analysis into the bias from the approximation (first step, only ERA5), and the bias between the two data set (ERA5 and RO), aiming to mitigate this specific problem (see new comment in Section 4.2).

*224: provide references to those missions.*

Note, we provided references in the paragraph from line 155 to 160.

*231-233: The alignment of an increase of systematic bias with the drop in the number of RO profiles is a very interesting feature. However, my question is how you can be certain that only this factor explains the increase of bias. No proof is provided, so the statement should be milder and speculative. From the statistical perspective, a reduction of the number of profiles would only increase the random error.*

Yes, we revised the sentence to a more conservative formulation. Furthermore, we cited the work of Scherllin-Pirscher et al. (2011) and Schwarz et al. (2017).

*Figure 7 is another proof, that v_eb (as well as v_g) are likely unable to approximate v_o.*

Please refer to our answer to major comment 5 on this.

*276-277: as this is not some new conclusion, I would say "as in Healy et al. (2020)".*

We mention now the important initial pre-work of Healy et al. (2020). Nevertheless, note that this work here is a more detailed analysis of the wind fields across the equator, providing further insight about the potential of RO data for wind field monitoring.

*277: what do you mean by "the resolution was possible to obtain" (I could not understand with going back to the results section). Please, express more clearly.*

Please see our answer to major comment 5 on this. We revised some statements of the final section.

*1: vertically*

*7-8: sentence "We analyze the equatorial balance equation within this latitude band." Is redundant in my opinion.*

*30: several heights but mostly upper troposphere*

*61: no comma before "isobaric levels".*

*63: between 15N in 10 S.*

*65: analyzed instead of "started to analyze"*

*66: to reproduce ERA5 geostrophic winds ("original winds" sound like total winds). You properly introduce "original" only later in the text, in line 115, leaving the reader confused at this stage.*

*68: I am not sure whether "Anthes" region is an established geographical term. Did you perhaps mean Andes?*

*70: equatorial band*

*70: "approaches" instead of "converges"*

*71: Reformulate sentence "Interesting was also to see…"*

*79: latitude-longitude*

*97-104: it is necessary to mention, that Coriolis parameter is now approximated using equatorial beta-plane approximation.*

*106: remove "still"*

*135: "includes"/"provides" instead of "combines"*

*135 and 137: sometimes you use "data" as singular noun and sometimes as plural, e.g. "The ERA5 reanalysis data combines…" vs. "The data are available…"*

*138: no comma before "to find", no comma before "for the equatorial balance…" Revise misuse of comma at the other places of the text as well.*

*178: no comma*

*Figure 4 should include two more rows: 1) monthly-mean winds V_o and 2) monthly-mean winds V_eb. The caption should be: Temporal development of the wind-speed bias…*

*222: revise "high occultation statistics"*

*237: "section"*

*237: no comma before "to complete"*

*244-245: "geostrophic break down" to the "the geostrophic approximation does not apply any more"*

*272: it is again unclear to the reader, what are the "winds calculated using ERA5 data and original winds". Try forming the Conclusions in a way that is understood even to readers who did not read the whole methodology*

*276: word order: "we could successfully apply"*

*287: first comma is excessive*

*279: this reads as the zonal wind speeds are 1 m/s and meridional wind speeds are 15 m/s. Again, be more precise for which levels in the equatorial +-5 deg channel do this wind speeds apply.*

Thank you for noticing! All corrections were implemented.

References listed by referee #2:

Baker, W. E., et al., 2014: Lidar-Measured Wind Profiles: The Missing Link in the Global Observing System. Bull. Amer. Meteor. Soc., 95, 543–564, https://doi.org/10.1175/BAMS-D-12-00164.1

Bonavita, 2023: On the limitations of data-driven weather forecasting models. ArXiv:2309.08473

Geer, A. J., Lonitz, K., Weston, P., et al., 2018: All-sky satellite data assimilation at operational weather forecasting centres. Quarterly Journal of the Royal Meteorological Society, 144, 1191–1217, https://doi.org/10.1002/qj.3202

Rennie, M. P., Isaksen, L., Weiler, F., de Kloe, J., Kanitz, T. & Reitebuch, O., 2021: The impact of Aeolus wind retrievals on ECMWF global weather forecasts. Quarterly Journal of the Royal Meteorological Society , 147, 3555–3586, https://doi.org/10.1002/qj.4142

Pourret, V., Šavli, M., Mahfouf, J.-F., Raspaud, D., Doerenbecher, A., Bénichou, H., et al., 2022: Operational assimilation of Aeolus winds in the Météo-France global NWP model ARPEGE. Quarterly Journal of the Royal Meteorological Society, 148, 2652–2671, https://doi.org/10.1002/qj.4329

Semane, N., R. Anthes, J. Sjoberg, S. Healy, and B. Ruston, 2022: Comparison of Desroziers and Three-Cornered Hat Methods for Estimating COSMIC-2 Bending Angle Uncertainties. J. Atmos. Oceanic Technol., 39, 929–939, https://doi.org/10.1175/JTECH-D-21-0175.1.

Zaplotnik, Ž., Žagar, N. & Semane, N., 2023: Flow-dependent wind extraction in strong-constraint 4D-Var. Quarterly Journal of the Royal Meteorological Society, 149, 2107–2124, https://doi.org/10.1002/qj.4497

---

## Referee Report (RR1)

J. Danzer, M. Pieler, and G. Kirchengast: "Closing the gap in the tropics: the added value of radio-occultation data for wind field monitoring across the equator", submitted to Atmospheric Measurement Techniques. Revised manuscript after first round of comments from reviewers.

The revised manuscript is substantially improved compared to the original version. The authors have carefully addressed the issues raised by both reviewers. I'm particularly pleased to see that they removed the comparison between ERA5- and RO-derived wind fields below 800 hPa.

Minor: Section 4.3, line 326, states that Fig. 8 shows data down to 1000 hPa. This should be 800 hPa.

I recommend publishing the manuscript after correction of this minor typograhical error.

---

## Referee Report (RR2)

Review of the study "Closing the gap in the tropics: the added value of radio-occultation data for wind field monitoring across the equator" by Danzer et al.

The authors have made substantial improvements to the manuscript and have properly addressed most of my comments. The purpose of this study is now nicely communicated, and the way if fills the scientific gap is described as well. Furthermore, the authors have performed extra analyses, which further confirm the robustness of their results. My suggestion would be to address better a single major point that has to be revised prior to acceptance. I think nothing is wrong with the analysis, but in the conclusions, the authors are overestimating the usefulness of GPSRO and equatorial wind balance to assess the meridional wind component. When this is revised, I would suggest accepting the paper.

**Major comments:**
It is clear from Figure 2 in in the manuscript that the equatorial wind balance is not applicable for the meridional wind in the tropical troposphere. The equatorial-balance bias is evidently above the threshold in many longitude bands below 100 hPa (Fig. 2c,f). This is more pronounced for the meridional wind, which can be seen by comparing Fig. 2d and Fig. 2e. Below, I attach some of my computations for ERA5, April 2010, 2.5x2.5 grid, for 10, 70, 200 and 500 hPa (see Figures 1-4). These plots confirm my suspicions that the equatorial wind balance is inappropriate for the estimation of meridional wind component, while the approximation works out nicely for the zonal wind component (the full fields and not only for the zonal-mean zonal wind as suggested in previous studies, which should be better communicated in this study).

In my opinion, it would be important to see not only the absolute error of the reconstruction (as in Fig 2f) but also the relative error, which often exceeds 100% (see attached Figures 5 and 6 below).

I do not fully understand, why Fig.3 compares $u_{eb}$ to $V_o$. In my opinion, $u_{eb}$ should be compared to $u_o$, $v_{eb}$ should be compared to $v_o$ and $V_{eb}$ should be compared to $V_o$, as in Figure 2. What is the aim of that?

The WMO threshold should not be followed blindly. The accurate description of climate trends of tropospheric meridional winds is extremely important as they describe the upper and lower branches of the Hadley circulation, which governs the precipitation distribution in the Tropics and Subtropics. The annual-mean magnitude of meridional wind in the upper branch of HC is around 1.5 m/s (Figure 0). In this respect, the WMO threshold is much too high.

[Figure]

**Figure 0.** Annual-mean meridional wind as a function of latitude and pressure in ERA5, 1979-2018.

Lines 8-10 and line 16 in the Abstract should therefore be revised – I am not convinced about the added value of meridional wind component for the reasons state above. I think the ability to reconstruct zonal winds (and not only the zonal-mean zonal winds) is still a nice result, but it has to be accurately communicated precisely both in the Abstract as well as in the Conclusions, as well as in the main text (e.g. discussion in lines 238-240).

[Figure]

**Figure 1.** ERA5 original wind [left] and equatorial wind balance reconstruction [right] for zonal wind [top] and meridional wind [bottom]. ERA5, April 2010, 10 hPa pressure level.

[Figure]

**Figure 2.** ERA5 original wind [left] and equatorial wind balanced reconstruction [right] for zonal wind [top] and meridional wind [bottom]. ERA5, April 2010, 70 hPa pressure level.

[Figure]

[Figure]

**Figure 4.** ERA5 original wind  [left] and equatorial wind balanced reconstruction [right] for zonal wind [top] and meridional wind [bottom]. ERA5, April 2010, 500 hPa pressure level.

[Figure]

**Figure 5.** Absolute errors [left] and relative errors [right] of equatorial wind balance reconstruction for zonal wind [top] and meridional wind [bottom]. ERA5, April 2010, 70 hPa pressure level.

[Figure]

**Figure 6.** Absolute errors [left] and relative errors [right] of equatorial wind balance reconstruction for zonal wind [top] and meridional wind [bottom]. ERA5, April 2010, 500 hPa pressure level.

[Figure]

**Figure 7.** Absolute errors [left] and relative errors [right] of equatorial wind balance reconstruction for zonal wind [top] and meridional wind [bottom]. ERA5, April 2010, 500 hPa pressure level. Note that the relative errors were now computed differently than in Figure 6, see figure caption.

**Minor comments:**

I currently do not see the potential of assimilating GPSRO derived winds for NWP/reanalyses applications, opposed to the author's response to reviewers. While the equatorial wind balance applies well particularly in the monthly mean fields in the stratosphere (Fig. 2), it is somewhat less applicable for the upper troposphere and below, and even less applicable for instantaneous fields. Another issue would be the observation correlation between the already assimilated bending angles and the GPSRO winds. On the other hand, I agree there is certainly great potential in the climate scope – both for monitoring as well as climate model verification, as it is also outlined in the paper. While the authors mention that they "do not focus on analysing or describing at the same time atmospheric dynamical processes as such; this is not within the scope of this study", they should note that the atmospheric dynamics are essential to the [potential of the] GPSRO wind retrieval.

Lines 5,6: Considering the previous comment, the equatorial balance approximation becomes important in the temporal mean, while the geostrophic balance mostly applies well for the fields at any time instance too. At any time instance, we do not have a predominant balance between winds and geopotential in the tropical upper-troposphere, the fields are only multivariately coupled through the equatorial modes (e.g. Matsuno, 1966).

25: vertical resolution of wind information (in the current form it might be misread as the vertical wind component)

34-37: reformulate the tenses. ADM Aeolus is down now. It also depends on the hydrometeor Mie scattering.

77: what observing system change do you refer to? Please, be more specific.

83: stratospheric zonal-mean wind field

102: The geostrophic balance breaks down in the tropics, due to the...

107: accuracy

109-110: Try to avoid "strong" and be more specific about the averaging, e.g. as reformulate as: "Since our focus are monthly-averaged mesoscale (might even be synoptic-scale already) winds relevant for the description of climate, ...
120: remove new paragraph indent

125: great! Very convincing!

138: equatorial balance approximation takes over. [not the winds themselves]

133 and Table 1: you now use the same subscript for globe and geostrophic. At this point, it seems somewhat puzzling.

Figure 1: is this based on the fields at single time instance or whole 2009, as suggested in line 149.

167: convincing!

174: consider "timeframe" → temporal averaging

175: reformulate

200: "results": which results, be more specific? Was the systematic data bias reduced?

220: Figure 2: do these statistics apply for time-mean data (year 2009) or for some specific time instance? It should be explained somewhere.

Figure 3: (a,b) and (c,d) have the same captions, despite first two representing absolute error and the second two the relative error.

238-239 I do not agree with the statement: "The difference fluctuates within he ±2 m s−1 threshold, also in the tropical troposphere." Also, such threshold is irrelevant given the small magnitude of the meridional wind.

242-244: Again, I do not agree with the reasoning here. The sole reason, why "it was possible to derive the wind fields close to the original wind speed" is because the zonal wind contributes the most to the wind speed and that component is well reconstructed by the equatorial wind balance.

252-257: I am not sure this improvement comes from the right source. The meridional winds could have wrong sign here and the wind speed would still improve. Any meridional wind addition would improve the total wind, i.e. if v<<u, adding v would bring the reconstructed total wind closer to the true total wind.

Fig 5f: the reconstruction error in f) is unacceptably large to capture the upper branch of the southern Hadley cell, the main feature of the tropospheric tropical circulation, despite fulfilling WMO requirements.

---

## Author Response (AR2)

**Author's Response to Referee #1 Comments**

J. Danzer, M. Pieler, and G. Kirchengast: "Closing the gap in the tropics: the added value of radio-occultation data for wind field monitoring across the equator", submitted to Atmospheric Measurement Techniques. Revised manuscript after first round of comments from reviewers.

The revised manuscript is substantially improved compared to the original version. The authors have carefully addressed the issues raised by both reviewers. I'm particularly pleased to see that they removed the comparison between ERA5- and RO-derived wind fields below 800 hPa.

Minor: Section 4.3, line 326, states that Fig. 8 shows data down to 1000 hPa. This should be 800 hPa.

I recommend publishing the manuscript after correction of this minor typograhical error.

We thank the reviewer very much for this positive general comment on our revised manuscript; we indeed had aimed to carefully improve in all points of concern and address all issues raised. Regarding the minor comment, we corrected the typo and are happy that you recommend the paper for publishing!

**Author's Response to Referee #2 Comments**

The authors have made substantial improvements to the manuscript and have properly addressed most of my comments. The purpose of this study is now nicely communicated, and the way if fills the scientific gap is described as well. Furthermore, the authors have performed extra analyses, which further confirm the robustness of their results. My suggestion would be to address better a single major point that has to be revised prior to acceptance. I think nothing is wrong with the analysis, but in the conclusions, the authors are overestimating the usefulness of GPSRO and equatorial wind balance to assess the meridional wind component. When this is revised, I would suggest accepting the paper.

We also thank this reviewer very much for this positive overall comment and basic acceptance suggestion, and for the constructive further input, especially on the "single major point". Regarding this one main point remaining (Major comment below), we feel that there possibly has been a partial misunderstanding of our results related to the co-benefit of the meridional component (possibly also since our explanation was in parts not clear and precise enough). We tried to clarify this now in our answer and at the same time improved relevant formulations in the further revised manuscript.

**Referee #2 (Major comment)**
**1)** It is clear from Figure 2 in in the manuscript that the equatorial wind balance is not applicable for the meridional wind in the tropical troposphere. The equatorial-balance bias is evidently above the threshold in many longitude bands below 100 hPa (Fig. 2c,f). This is more pronounced for the meridional wind, which can be seen

by comparing Fig. 2d and Fig. 2e. Below, I attach some of my computations for ERA5, April 2010, 2.5x2.5 grid, for 10, 70, 200 and 500 hPa (see Figures 1-4). These plots confirm my suspicions that the equatorial wind balance is inappropriate for the estimation of meridional wind component, while the approximation works out nicely for the zonal wind component (the full fields and not only for the zonal-mean zonal wind as suggested in previous studies, which should be better communicated in this study).

In my opinion, it would be important to see not only the absolute error of the reconstruction (as in Fig 2f) but also the relative error, which often exceeds 100% (see attached Figures 5 and 6 below). I do not fully understand, why Fig.3 compares u_eb to V_o. In my opinion, u_eb should be compared to u_o, v_eb should be compared to v_o and V_eb should be compared to V_o, as in Figure 2. What is the aim of that? The WMO threshold should not be followed blindly. The accurate description of climate trends of tropospheric meridional winds is extremely important as they describe the upper and lower branches of the Hadley circulation, which governs the precipitation distribution in the Tropics and Subtropics. The annual-mean magnitude of meridional wind in the upper branch of HC is around 1.5 m/s (Figure 0). In this respect, the WMO threshold is much too high.

Lines 8-10 and line 16 in the Abstract should therefore be revised – I am not convinced about the added value of meridional wind component for the reasons state above. I think the ability to reconstruct zonal winds (and not only the zonal-mean zonal winds) is still a nice result, but it has to be accurately communicated precisely both in the Abstract as well as in the Conclusions, as well as in the main text (e.g. discussion in lines 238-240).

Thank you for these thoughtful considerations and also the complementary ERA5 figures. We like to first of all clarify that we did not (aim to) make the statement in the manuscript that the meridional wind component itself is well reconstructed in the troposphere. Instead, we stated that the (zonal-mean) total wind speed (in the troposphere) benefits from computing both components, compared to when just using the zonal component as an approximation for the total wind speed. This can be seen, for example, in Figure panels 3b and 3d compared to panels 3a and 3c. And, furthermore, in Figure panels 6g when comparing to 6d, at the 200 hPa level. The zonal-mean total wind speed in the troposphere is always somewhat better approximated when including both, the zonal and meridional wind components. We hence respectfully think that related to this aspect, there was possibly a misunderstanding (possibly also since our text was not fully clear and precise enough to this end). We think this co-benefit aspect of the meridional component (while in itself it is not of sufficient utility) is also a quite interesting result and hence prefer to keep it as one of the messages of this paper.

To help avoid potential misunderstanding, we carefully rechecked the text in the manuscript, and improved at several places, to ensure that we now clearly express this specific kind of co-benefit for the zonal-mean total wind speed in the troposphere (while in the stratosphere, where the absolute values of meridional wind component are very small, also the added value is not there). We also eliminated to show the WMO target requirement in a relevant figure panel (Figure 2f) as if it would apply also to the meridional component on its own; we agree this was a bit sloppy, since this requirement is meant to apply to the total wind speed (and likewise the zonal wind component which strongly dominates the total wind speed).

Furthermore, we understand that the reviewer suggests additional visualization of relative errors. However, we feel we already show relative errors to the point we consider needed, which is within Figure 3 in the manuscript. This sufficiently supports the one key result explained just before, i.e., supports the message that the zonal-mean total wind speed is better approximated in the troposphere by including both wind components. One can clearly see the wind speed bias improvements in Figure 3b (absolute difference) and 3d (relative difference), throughout the equatorial troposphere. While in principle we could hence add a further relative difference plot, we consider that this does not add further value to the relevant key messages of this study. Rather than added visualization, we of course tried to carefully improve the related explanations so as to avoid any misunderstanding of the value of the meridional wind component (just the mentioned co-benefit), since nearly all of the wind speed information (and essentially all of the longitudinally resolved one) is indeed carried by the zonal wind component.

Overall, from the line of arguments above, we included in the further revised manuscript the following main improvements:

1. We more strongly emphasized and better clarified in the manuscript, such as in the abstract, the discussion of results, and the conclusions, that while the meridional component itself is not well estimated, it is the zonal-mean total wind speed in the troposphere that benefits from including both components. We also eliminated to indicate the WMO target requirements in a panel showing meridional wind only.
2. We more clearly stated that the equatorial balance approximation works best in the stratosphere (abstract and conclusions). That is indeed a very valuable constructively critical point of the reviewer. We should have emphasized this more strongly before.
3. We preferred not to include another relative difference figure, since it would not add to our message, but we rather aimed to bring the message better along with better discussion especially of Figure 3.

**Referee #2 (Minor comments)**

I currently do not see the potential of assimilating GPSRO derived winds for NWP/reanalyses applications, opposed to the author's response to reviewers. While the equatorial wind balance applies well particularly in the monthly mean fields in the stratosphere (Fig. 2), it is somewhat less applicable for the upper troposphere and below, and even less applicable for instantaneous fields. Another issue would be the observation correlation between the already assimilated bending angles and the GPSRO winds. On the other hand, I agree there is certainly great potential in the climate scope – both for monitoring as well as climate model verification, as it is also outlined in the paper. While the authors mention that they "do not focus on analysing or describing at the same time atmospheric dynamical processes as such; this is not within the scope of this study", they should note that the atmospheric dynamics are essential to the [potential of the] GPSRO wind retrieval.
Lines 5,6: Considering the previous comment, the equatorial balance approximation becomes important in the temporal mean, while the geostrophic balance mostly applies well for the fields at any time instance too. At any time instance, we do not have a predominant balance between winds and geopotential in the tropical upper-troposphere, the fields are only multivariately coupled through the equatorial modes (e.g. Matsuno, 1966).

Thanks for this comment. However, we want to emphasize that we do not discuss or suggest in this paper the potential to assimilate RO-derived winds in NWP. We rather fully agree with the reviewer that it would be highly unusual (not to say highly suboptimal to useless) to assimilate a derived diagnostic quantity like the RO-derived winds. That is, of course the RO's information content is best made available to data assimilation and forecasting systems by using quite more low-level data such as the well-proven assimilation of RO bending angles (as also done in the ECMWF IFS). In this paper, we purely focus on and discuss the climate-related wind field retrieval, for the benefit of long-term stable climate wind field monitoring.

25: vertical resolution of wind information (in the current form it might be misread as the vertical wind component)

Thank you, we implemented this improvement.

34-37: reformulate the tenses. ADM Aeolus is down now. It also depends on the hydrometeor Mie scattering.

We rephrased to:
"On the other hand, the Atmospheric Dynamics Mission (ADM-Aeolus, operating over August 2018 to July 2023) provided 3D wind profiling with a frequent and high-resolution coverage, filling measurement gaps over the oceans, poles, tropics, and the southern hemisphere, up to an altitude of about 20 km. However, it depended on clear-air molecular scattering (no measurements within clouds) and on hydrometeor Mie scattering, which can

be particularly tricky at tropical latitudes, due to the high-altitude cloud systems (see also Stoffelen et al., 2005, 2020; Kanitz et al., 2019). "

77: what observing system change do you refer to? Please, be more specific.

We rephrased to:
"Our study furthermore showed that within the 2007 to 2020 evaluation period the difference between RO and ERA5 became noticeably smaller from 2016 onward, coinciding with an ERA5 observing systems change including as of 2016 additional information from various sources such as land stations, ships, and buoys."

83: stratospheric zonal-mean wind field
102: The geostrophic balance breaks down in the tropics, due to the…
107: accuracy

Ok, done.

109-110: Try to avoid "strong" and be more specific about the averaging, e.g. as reformulate as: "Since our focus are monthly-averaged mesoscale (might even be synoptic-scale already) winds relevant for the description of climate, …
120: remove new paragraph indent
125: great! Very convincing!
138: equatorial balance approximation takes over. [not the winds themselves]

Ok, thank you, done.

133 and Table 1: you now use the same subscript for globe and geostrophic. At this point, it seems somewhat puzzling.

We are sorry for having had this confusion. The subscript (g) still refers to the geostrophic approximations, which was studied on the complete globe. The description in Table 1 is now clarified by changing to:
"(eb): focus area ±5◦ N/S; (g): studied on complete globe"

Figure 1: is this based on the fields at single time instance or whole 2009, as suggested in line 149.

The figure shows the results for January 2009. We included this information now in the manuscript.

167: convincing!
174: consider "timeframe" à temporal averaging
175: reformulate

Thank you, done.

200: "results": which results, be more specific? Was the systematic data bias reduced?

Thank you for pointing to this, we reformulated to:
"Tests revealed that a Gaussian smoothing with a 5° longitudinal smoothing window improved the wind data estimation and the systematic difference decreased."

220: Figure 2: do these statistics apply for time-mean data (year 2009) or for some specific time instance? It should be explained somewhere.

For January 2009, as the text states in line 222. We also added this now in the description of Figure 2.

Figure 3: (a,b) and (c,d) have the same captions, despite first two representing absolute error and the second two the relative error.

Thank you for noticing this. We updated the Figure 3 accordingly.

238-239 I do not agree with the statement: "The difference fluctuates within he ±2 m s−1 threshold, also in the tropical troposphere." Also, such threshold is irrelevant given the small magnitude of the meridional wind.

Thank you for this comment. And indeed, regarding the "threshold", we think you are right and we have been a bit sloppy to this end to show it also for meridional wind. We agree it is sound to show this indicative target requirement of the WMO only for the total wind speed and the zonal wind component (that strongly dominates the total wind speed) but not for the small meridional wind component. For this reason, we have removed the target requirement indication from Figure 2f and made sure, at necessary places, to revise the text accordingly to avoid wording relating the meridional wind to this requirement.

Along with this, we deleted and reformulated the particular statement to:
"Analyzing the equatorial-balance bias shows that the difference fluctuates with amplitudes of about ±2 to 4 m/s in the tropical troposphere (Figure 2f)."

242-244: Again, I do not agree with the reasoning here. The sole reason, why "it was possible to derive the wind fields close to the original wind speed" is because the zonal wind contributes the most to the wind speed and that component is well reconstructed by the equatorial wind balance.
252-257: I am not sure this improvement comes from the right source. The meridional winds could have wrong sign here and the wind speed would still improve. Any meridional wind addition would improve the total wind, i.e. if v<<u, adding v would bring the reconstructed total wind closer to the true total wind.

Please see our answer to the major concern above, related to these arguments. We tried to clarify there, and we further emphasized in the revised manuscript more strongly that while the meridional component itself is not well estimated, it is the zonal-mean total wind speed in the troposphere that benefits from including both components. We reformulated at several places in the manuscript.
Such as:
"This result indicates that while the meridional component itself is not well estimated, the calculation of zonal-mean total wind speed benefits from including the meridional wind component in the troposphere, since it brings the reconstructed wind speed closer to the original wind. However, in the stratosphere, the close-to-zero meridional wind brings in no added value."
As to the point of "right source" for improvement, we think the key is that we now better clarify that the improvement plays out in the zonal-mean wind speed in the troposphere (i.e., not the longitudinally resolved one); and yes, the overall beneficial effect in this zonal-mean wind speed is evidently coming up from the zonal wind component alone providing a little underestimation of the speed (and also yes, we won't consider the data viable to reconstruct a wind direction).

Fig 5f: the reconstruction error in f) is unacceptably large to capture the upper branch of the southern Hadley cell, the main feature of the tropospheric tropical circulation, despite fulfilling WMO requirements.

Yes, we agree and carefully revised the manuscript, such as:
"When considering the differences between the two data sets for meridional wind and wind speed (Figure 5f and 5i) it is seen that these also generally reside within ±2 ms$^{-1}$. Nevertheless, due to the already small absolute

magnitudes of the meridional wind (Figure 5d and 5e) it is also clear that this component itself is not well reproduced. However, the zonal-mean total wind speed (bottom row) still benefits in the troposphere from including both wind components (Figure 5i), with the dominant contribution coming from the zonal component."

And also, in lines 305 to 308:
"Summarizing the results of the current and previous section, meso-scale climate wind field derivation was possible across the equator using RO data, when focusing on its core vertical region of high quality and resolution. Furthermore, we found that while the meridional component itself is not well estimated, it is the zonal-mean total wind speed in the troposphere that benefits from including both components, while in the stratosphere the meridional component's influence becomes negligible."

---

## Author Response (AR3)

**Comment Editor:**

The updated manuscript has addressed the issues raised by the Reviewer. However, conclusions regarding the "added value" of including the meridional component in the calculation of the total speed should be more clearly explained.

**Answer Authors:**

Dear Editor, thank you for stating that the authors have essentially addressed all issues, leaving only one final point as a further clarification, which regards the "added value of including the meridional component v in the estimation of the total wind speed V", and asked for a comment back to you.

Our comment to this final point is:

We had already included an explanation to this end in the previous response to the Reviewer 2 but confirm, we did not bring this more specific explanation into the manuscript. We note that, related to the above statement, we have also checked for all 2.5° x 2.5° grid points within 5°S to 5°N and over all longitudes, how many grid points have the direction of v_o (original v-component) and v_eb (approximated by the eq. balance approximation) aligned (="same sign") vs how many don't agree ("opposite sign"). The result was that the approximation is not fully uninformative related to the v-component direction: we obtained in the mean over all 120 hPa to 800 hPa pressure levels, and over all twelve months of our 'test year' 2009, roughly ~55-60% same-sign grid points, but still in average >40% opposite sign. Hence it is somehow true that, while the (small) magnitude of the v-component is modeled reasonably, helping to compensate the slight underestimation of V_eb (the total wind speed) by the u_eb component alone in a good way [e.g, as Figure panels 2b,d vs. 2a,c show], the direction is not fully but mostly "noise-like", and therefore not viable to reconstruct a wind direction.

Based on this explanation, we weakened at some text places the wording of "benefit" or "added value" and added the following improvements in the manuscript text:

Line 10:

*"However, we still found a somewhat better agreement from including both components in the zonal-mean total wind speed in the troposphere."*

Line 16:

*"Overall, the study encourages the use of RO wind fields for meso-scale climate monitoring over the entire globe, including the equatorial region, and also showed a small improvement in the troposphere when including the meridional wind component in the zonal-mean total wind speed."*

Line 261:

*"This result indicates that while the meridional component itself is not well estimated, the calculation of the zonal-mean total wind speed shows a somewhat closer agreement from including also this small wind component in the troposphere. That is, the slight underestimation left by the zonal wind speed in the troposphere (see Fig. 3a,c) is mitigated by the inclusion of the meridional wind speed, which brings the approximated zonal-mean total wind speed closer to the original one (see Fig. 3b,d). The reason for this small co-benefit to the zonal-mean total wind speed is that the averaged meridional wind speed is estimated by the balance approximation at about the right (small) magnitude in the troposphere. However, in the stratosphere, the close-to-zero meridional wind brings in no further improvement. Nevertheless, we caution that, in addition to finding the meridional wind estimates not viable to benefit*

*also longitude-resolved wind speeds, we also find them not viable to reconstruct wind direction (i.e., wind vectors on top of wind speed estimates). The reason is that the direction of the approximated meridional wind at individual grid points is nearly as often found opposite as it is found aligned with the original wind direction."*

Line 366:

*"Hence, a wind flow with small magnitudes and also changes in the direction of the flow (changing sign) is challenging to reproduce. We find in this respect the equatorial balance approximation not suitable for the reconstruction of wind directions and of longitude-resolved wind speeds in the troposphere."*

Line 384:

*"To summarize, we found encouraging results in that we revealed that RO data do indeed have good potential for long-term wind field monitoring over the complete globe, including across the equator. A meso-scale climate resolution (5° x 5° latitude x longitude grid) was possible to be demonstrated for the RO data in this specific region for wind speed, with evidence for added value from their accuracy and high resolution, as well as their long-term stability."*